# From single neurons to behavior in the jellyfish *Aurelia aurita*

Fabian Pallasdies*, Sven Goedeke, Wilhelm Braun, Raoul-Martin Memmesheimer*

Neural Network Dynamics and Computation, Institute of Genetics, University of Bonn, Bonn, Germany

**Abstract** Jellyfish nerve nets provide insight into the origins of nervous systems, as both their taxonomic position and their evolutionary age imply that jellyfish resemble some of the earliest neuron-bearing, actively-swimming animals. Here, we develop the first neuronal network model for the nerve nets of jellyfish. Specifically, we focus on the moon jelly *Aurelia aurita* and the control of its energy-efficient swimming motion. The proposed single neuron model disentangles the contributions of different currents to a spike. The network model identifies factors ensuring non-pathological activity and suggests an optimization for the transmission of signals. After modeling the jellyfish's muscle system and its bell in a hydrodynamic environment, we explore the swimming elicited by neural activity. We find that different delays between nerve net activations lead to well-controlled, differently directed movements. Our model bridges the scales from single neurons to behavior, allowing for a comprehensive understanding of jellyfish neural control of locomotion.

## Introduction

### Modeling jellyfish

Understanding how neural activity leads to behavior in animals is a central goal in neuroscience. Since jellyfish are anatomically relatively simple animals with a limited behavioral repertoire (*Albert, 2011*), modeling their nervous system opens up the possibility to achieve this goal.

Cnidarians (in particular jellyfish) and ctenophores (comb jellies) are the only non-bilaterian animal phyla with neurons. While their phylogenetic position is still not entirely resolved, evidence suggests that cnidarians are our most distant relatives with homologous neurons and muscles (*Steinmetz et al., 2012*; *Marlow and Arendt, 2014*; *Moroz and Kohn, 2016*). Well-preserved fossils of medusozoa from the Cambrian (*Cartwright et al., 2007*) and evidence for medusoid forms from the Ediacaran (*Van Iten et al., 2006*) indicate that jellyfish are evolutionary old. These findings and their anatomical simplicity suggest that they are similar to the earliest neuron-bearing, actively swimming animals. Their study should therefore yield insight into the earliest nervous systems and behaviors.

The present study focuses on the neuro-muscular control of the swimming motion in a true (scyphozoan) jellyfish in the medusa stage of development. Specifically, incorporating available experimental observations and measurements, we develop a bottom-up multi-scale computational model of the nerve nets and couple their activity to a muscle system and a model of the bell of the moon jelly *Aurelia aurita* (see *Figure 1* for an overview of the model). Using fluid-structure hydrodynamics simulations, we then explore how the nervous system generates and shapes different swimming motions.

Before presenting our results, we review the current knowledge on the nervous systems of jellyfish in the following introductory sections, also highlighting specific open questions that further motivate our study.

**\*For correspondence:**
fabianpallasdies@gmail.com (FP);
rm.memmesheimer@uni-bonn.de
(RMM)

**Competing interests:** The authors declare that no competing interests exist.

**Figure 1.** Jellyfish anatomy and schematic overview of our model. (**A**) Moon jelly *Aurelia aurita* in water. The bell is nearly relaxed. Rhopalia are clearly visible as bright spots on wedge-shaped sections of the bell margin. The location of modeled nerve nets and muscles is marked by arrows. (**B**) Diagram of the jellyfish model components. Rhopalia can be excited by external stimulation. They are connected to both the motor nerve net (MNN) and the separate diffuse nerve net (DNN) on the subumbrella. The MNN selectively innervates the circular muscles, while the DNN selectively innervates the radial ones. The muscles deform the jellyfish bell, which interacts with the surrounding fluid. The muscle forces in turn depend on the bell shape. A putative coupling from MNN and DNN back to the rhopalia (not modeled in this study) is indicated by thin dashed arrows.

## Nervous systems of scyphomedusae

The nervous system of scyphozoan jellyfish consists of several neuronal networks, which are distributed over the entire jellyfish bell, the tentacles and the endoderm (*Schäfer, 1878*; *Passano and Passano, 1971*). The only obvious points of concentration of a larger number of neurons are the rhopalia (*Figure 1*), small sensory structures of which there are usually eight distributed around the margin of the bell (*Nakanishi et al., 2009*).

Much of the current knowledge on the inner workings of these nerve nets, in particular concerning the control of the swim musculature, was already formulated by George Romanes in the 19th century (*Romanes, 1885*). During the swimming motion almost all the subumbrellar muscles contract synchronously and push the jellyfish forward. In a series of cutting experiments, Romanes destroyed and removed parts of the umbrella. He found that the contraction usually starts at one of the rhopalia and propagates around almost arbitrarily placed cuts in the subumbrella. Furthermore, Romanes observed two different types of contraction waves: a fast, strong wave leading to the regular swimming motion and a slower wave, at about half the speed, which was so weak that one could hardly see it activate the swim musculature. When a slow contraction wave originating somewhere on the outer margin of the jellyfish umbrella reached a rhopalium, a fast excitation wave emerged from that rhopalium after a short delay.

With advancing neurobiological methods, Romanes' observations were later verified and expanded (*Passano, 1965*; *Satterlie, 2002*). This led to the identification of two different nerve nets, the motor nerve net (MNN) and the diffuse nerve net (DNN), which are responsible for the fast and slow contraction wave, respectively.

## The motor nerve net

The motor nerve net extends over the subumbrella (*Figure 1*) and consists of large neurons with usually two neurites (*Schäfer, 1878*; *Anderson and Schwab, 1981*; *Satterlie, 2002*). The neurons function in basically the same manner as neurons with chemical synapses in higher animals (*Anderson and Schwab, 1983*; *Anderson, 1985*).

The MNN is through-conducting in the sense that if a small number of neurons is activated, a wave of activation spreads over the entire network, leading to a series of neuronal discharges. The conduction speed is between 45 cm/s and 1 m/s (*Horridge, 1956*; *Passano, 1965*). The activation is preserved even if large parts of the network are destroyed. It generates Romanes' fast contracting wave in the swim musculature (*Horridge, 1956*).

Spontaneous waves in the MNN are initiated by pacemakers located in each of the rhopalia (*Passano, 1965*). After firing, the wave-initiating pacemaker resets and the other ones reset due to the arriving MNN activity. *Horridge (1959)* showed that sensory input modulates the pacemaker activity. This may be one of the main mechanisms of sensory integration and creation of controlled motor output in the jellyfish.

In studies that investigated the electrophysiology of the MNN in detail, remarkable features have been observed. First, even though the synapses seem to be exclusively chemical they are symmetrical, both morphologically and functionally. Both sides of the synaptic cleft have a similar structure containing vesicles as well as receptors (*Horridge and Mackay, 1962*). In particular, the neurites do not differentiate into axon and dendrite. *Anderson (1985)* directly showed that the conduction is bidirectional. Electrical synapses have not been found, neither through staining nor in electrophysiological experiments (*Anderson and Schwab, 1981*; *Anderson, 1985*; *Anderson and Spencer, 1989*). Second, synapses are strong, usually creating an excitatory postsynaptic potential (EPSP) that induces an action potential (AP) in the receiving neuron (*Anderson, 1985*).

This fits with the observation that the MNN remains robustly through-conducting during cutting experiments (*Horridge, 1954b*). However, it also raises the question why symmetrical synapses of such strength do not lead to repetitive firing in (sub-)networks of neurons or even to epileptic dynamics.

## The diffuse nerve net

Historically, any neuron not associated with the MNN or the rhopalia was categorized into the DNN, including the neurons in the manubrium and the tentacles (*Horridge, 1956*). We adopt the nomenclature of more recent studies, where the term DNN refers mostly to the through-conducting nerve net of the ex- and subumbrella, which does not directly interact with the MNN (*Figure 1*) (*Arai, 1997*). Little is known about the DNN's small neurons and its synapses. The conduction speed of activity waves (15 cm/s) along the subumbrella is less than in the MNN (*Passano, 1973*).

*Horridge (1956)* was the first to suggest that innervation of the swim musculature via the DNN with its slower time scale may allow for a different activation pattern and thereby induce a turning motion. This could be achieved by a simultaneous versus a successive arrival of MNN- and DNN-

generated contraction waves on two sides of the animal. In *Aurelia*, however, no visible contraction of the regular swim musculature after DNN excitation was observed (*Horridge, 1956*). Still, the DNN might influence the circular muscles by amplifying the impact of the MNN activity as it was measured in other jellyfish (*Passano, 1965*; *Passano, 1973*). In addition, there is a small band of radial muscles on the marginal angles of *Aurelia*, which contract during a turning motion (*Gemmell et al., 2015*). The speed of the muscle activation and the position of the muscles indicate that they are innervated by the DNN.

In accordance with the idea of a coupled activation of DNN and MNN, DNN activity can activate the MNN indirectly via a rhopalium. The delay observed between DNN activity arrival and the initiation of the MNN activation is highly variable (*Passano, 1965*; *Passano, 1973*). Apart from this, the DNN does not directly interact with the MNN (*Horridge, 1956*). Some behavioral (*Horridge, 1956*; *Gemmell et al., 2015*) and anatomical (*Nakanishi et al., 2009*) evidence suggests that a rhopalium might activate the DNN together with the MNN in response to a strong sensory stimulus. These points indicate that each rhopalium is responsible for steering the animal by stimulating either one or both of the nerve nets. If and how the jellyfish can control its swimming motion beyond this is currently unknown.

## Hydrodynamics of swimming

Oblate-shaped jellyfish like *Aurelia* are among the most efficient swimmers in the world. Their cost of transport (energy consumption during movement per mass and movement distance) is very low (*Gemmell et al., 2013*). Therefore, there has been a continuous effort to understand the hydrodynamics of their swimming motion.

As described above, the jellyfish swim musculature is located solely on the subumbrella. Jellyfish do not have muscles that actively open the bell after a contraction. Instead their body is filled with mesoglea, a mixture of fluid and elastic fibers that create a hydrostatic skeleton. During a contraction the mesoglea stores elastic energy created by pushing the fluid to the center and stretching the fibers, which leads to relaxation of the bell when the muscle tension drops (*Alexander, 1964*; *Gladfelter, 1972*; *Gladfelter, 1973*).

The specific swimming mechanism of oblate jellyfish has been described as 'rowing' or 'paddling', as opposed to 'jetting', which is found in prolate jellyfish (*Colin and Costello, 2002*; *Sahin et al., 2009*). Jellyfish that use the latter swimming mechanism produce most of their forward momentum during their contraction phase, and get pushed forward by propelling fluid out of their bell (*Villanueva et al., 2010*). In contrast rowers produce their forward momentum through a series of vortex rings at the bell margin. Since these vortices form both during the contraction and the relaxation of the bell, rowers are highly cost efficient swimmers (*Colin and Costello, 2002*; *Dabiri et al., 2005*; *Dabiri et al., 2007*; *Gemmell et al., 2013*; *Gemmell et al., 2015*).

An important part of the insight into the swimming motion of animals has been gained through fluid dynamics simulations. Methods like the Immersed Boundary (IB) method have been applied to study the interactions of aquatic animals with the surrounding fluid (*Fauci and Peskin, 1988*; *Peskin, 2002*; *Cortez et al., 2004*; *Bhalla et al., 2013*). This revealed for example that in anguilliform swimmers, the same muscle activation patterns can produce different swimming motions depending on body stiffness (*Tytell and Lauder, 2004*; *Tytell et al., 2010*). Studies adopting an integrated view of neural circuitry and biomechanics (*Tytell et al., 2011*) developed closed-loop models for vertebrate swimmers, in which a central pattern generator circuit controls muscle activity interacting via the body shape with the surrounding fluid (*Ekeberg and Grillner, 1999*; *Hamlet et al., 2018*). *Herschlag and Miller (2011)*, *Park et al. (2014)* and *Hoover and Miller (2015)* used the IB method to simulate jellyfish motion in a fluid by modeling it as an immersed mechanical structure of springs and beams. *Herschlag and Miller (2011)* generated realistic jellyfish forward motion in 2D using a simple model of the bell kinematics. A related study, *Park et al. (2014)*, focused on the vortex formation during swimming. *Hoover and Miller (2015)* drove the bell of their model jellyfish at different frequencies. They found that frequencies around resonance, whose precise values depend on the contraction forces, are optimal for swimming speed and cost of transport.

Few studies have so far attempted to pin down the mechanisms of directional steering in jellyfish locomotion. Jellyfish turn by creating an asymmetric bell contraction (*Gladfelter, 1973*). In most scyphozoan jellyfish, the part of the bell on the inner side of the turn contracts stronger and earlier

(*Gladfelter, 1973*; *Horridge, 1956*). Horridge suggested that this is because the activities of DNN and MNN coincide there. Jellyfish often use this turning to adjust their tilt. The contraction wave then usually starts at the rhopalium on the inside of the turn (*Shanks and Graham, 1987*; *Horridge, 1956*). To our knowledge, *Hoover (2015)* contains the so far only modeling study on turning in jellyfish. Hoover created a 3D model of a jellyfish and tested the effect of a rectangular region of increased tension traveling in both directions around the bell. He found that the bell turns toward the direction of the origin of this traveling wave, as observed in real jellyfish. The amount of angular displacement depends strongly on the speed at which the activity travels around the bell.

Another component that is considered important for the swimming of jellyfish are the bell margins. During regular swimming, the margins of *Aurelia* are very flexible and follow the rest of the bell as it contracts and expands (*McHenry and Jed, 2003*). Robot and 3D models show that such 'flaps' enhance the performance of swimming by increasing the vorticity of the vortex rings that are shed off (*Colin et al., 2012*; *Villanueva et al., 2014*; *Hoover et al., 2017*). As described above, the bell margins in *Aurelia* do not possess circular muscles but rather a set of loosely organized radial ones (*Figure 1*). During turning maneuvers they stiffen the margins, starting at the origin of the activation wave (*Gemmell et al., 2015*). This, together with the observation that DNN activation creates no visible contraction of the circular muscles in *Aurelia* (*Horridge, 1956*), suggests that MNN and DNN each control one set of muscles and that this enables steering of the jellyfish. However, a mechanistic understanding how the activity of the two nerve nets determines turning is lacking. Furthermore, since the origin of nerve net activation waves is near the stimulus and apparently defines the inside of the turn, the hypothesis might only explain steering toward a stimulus. Some observations in jellyfish, for example their ability to keep a certain distance from rock walls (*Albert, 2008*; *Albert, 2011*), may, however, suggest that jellyfish are capable of steering away from aversive stimuli. It is currently unknown how the through-conducting nerve nets could allow such a level of control.

## Results

### A model for scyphozoan neurons

#### Model construction and comparison to data

We develop a biophysically plausible scyphozoan neuron model on the level of abstraction of Hodgkin-Huxley type single compartment models. These describe the actual voltage and current dynamics well and there is sufficiently detailed electrophysiological data available to fit such a model, obtained from *Cyanea capillata* (*Anderson, 1989*). Furthermore, dynamical mechanisms are not obscured by the presence of too many variables and the models lend themselves to fast simulations of medium size neural networks, with several thousands of neurons.

We incorporate the voltage-dependent transmembrane currents observed for scyphozoan MNN neurons by *Anderson (1987)* and *Anderson (1989)* and fit the model parameters to the voltage-clamp data presented there (see Materials and methods). The results of the fitting procedure are shown in *Figure 2A*. The current traces of the biophysical model agree well with the measured ones, both qualitatively and quantitatively, for the broad experimentally explored range of clamping from −20 mV to +90 mV (step-size: 7.5 mV, resting potential: −70 mV). The remaining unknown features of the model are the membrane capacitance and the synapse model. We choose them such that (i) the excitatory postsynaptic potentials resemble in their shape the experimentally found ones (*Anderson, 1985*), (ii) the inflection point of an AP is close to 0 mV (*Anderson and Schwab, 1983*) and (iii) it takes approximately 2.5 ms for an AP to reach peak amplitude after stimulation via an excitatory postsynaptic current (EPSC) (see *Figure 2B*) (*Anderson, 1989*).

#### Action potentials and synapses

Our model generates APs similar to the ones observed experimentally by *Anderson and Schwab (1983)*. It allows to quantitatively disentangle the contributions of the different transmembrane channel populations, see *Figure 2*. Before an AP, the leak current dominates. After the voltage surpasses the inflection point, the fast transient in- and outward currents generate the voltage spike. During the spike, the steady-state outward current activates and stays active during repolarization. The slow outward current does not activate, since it requires depolarizations beyond +55 mV (*Anderson, 1989*).

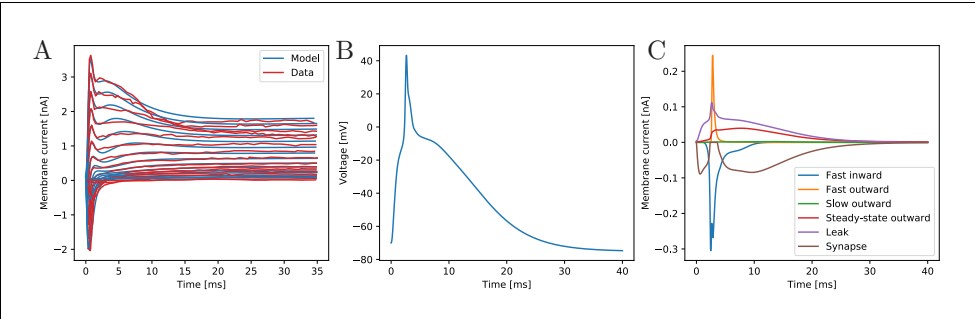

**Figure 2.** The biophysical model fitted to the voltage-clamp data. (**A**) Comparison of our model dynamics with the voltage-clamp data (*Anderson, 1989*) that was used to fit its current parameters. The model follows the experimentally found traces. (**B**) Membrane voltage of a neuron that is stimulated by a synaptic EPSC at time zero. The model neuron generates an action potential similar in shape to experimentally observed ones. (**C**) The disentangled transmembrane currents during an action potential.

As experimentally observed in scyphozoan MNN neurons (*Anderson, 1985*), our model EPSCs have fast initial rise, initially fast and subsequently slow decay and a single EPSC suffices to evoke an AP in a resting neuron (*Anderson, 1985*). Furthermore, we incorporate the experimentally observed synaptic rectification: the synaptic current influx decays to zero when the voltage approaches the reversal potential (+4 mV) but does not reverse beyond (cf. brown trace in *Figure 2C*). Synaptic transmission is activated when a neuron reaches +20 mV from below, which happens during spikes only. Since synapses in MNN neurons are symmetrical (*Anderson, 1985*; *Anderson and Grünert, 1988*), we hypothesize that after transmitter release into the synaptic cleft, both pre- and postsynaptic neurons receive an EPSC. In our model, this 'synaptic reflux' is responsible for a delayed repolarisation: the voltage stays near zero for several milliseconds after the fast return from the spike peak, see *Figure 2B*. This is also visible in electrophysiological recordings (*Anderson and Schwab, 1983*; *Anderson, 1985*).

## Refractory period

As a single AP evokes an AP in a resting postsynaptic neuron and synapses are bidirectional, one might expect that the postsynaptic AP (or even the reflux) in turn evokes further presynaptic APs. However, experiments in two-neuron systems do not observe such repetitive firing but only bumps of depolarization after a spike (*Anderson, 1985*). This is likely due to the long refractory period of scyphozoan neurons, which is initially absolute for about 30 ms and thereafter relative for about 70 ms (*Anderson and Schwab, 1983*). In agreement with experimental findings, we do not observe repetitive firing in systems of two synaptically connected model neurons, but only bumps of depolarization after a spike. This indicates that our model neurons have a sufficiently long refractory period, although it has not been explicitly inserted. *Figure 3A* shows as an example the voltage trace of a neuron that is stimulated by an EPSC, spikes and receives an EPSC due to the spiking of a postsynaptic neuron. Due to signal transmission delays, the neuron receives the second EPSC 7 ms after the first one.

To determine the refractory period effective under arrival of synaptic inputs, we apply two EPSCs with increasing temporal distance (see *Figure 3C*). We find a refractory period of about 20 ms. The longer refractory periods observed in scyphozoan neurons may be due to additional channel features that are not detectable from the voltage clamp data, such as delayed recovery from inactivation (*Kuo and Bean, 1994*; *French et al., 2016*). The synaptic and AP traveling delay in our model (at most 3.5 ms, see Materials and methods) plus the time to reach threshold (about 2.5 ms) are far from sufficient for the presynaptic neuron to recover from its spike, such that repetitive spiking is prevented, as observed in experiments.

To understand the origin of the effective refractory period's long duration, we determine it also in deficient model neurons, where the slow steady-state channel, the synaptic reflux and/or the synaptic rectifier (*Anderson, 1985*) are missing (*Figure 3C*). We find that the synaptic reflux and the steady-state current are crucial for the long duration: without them the refractory period is reduced

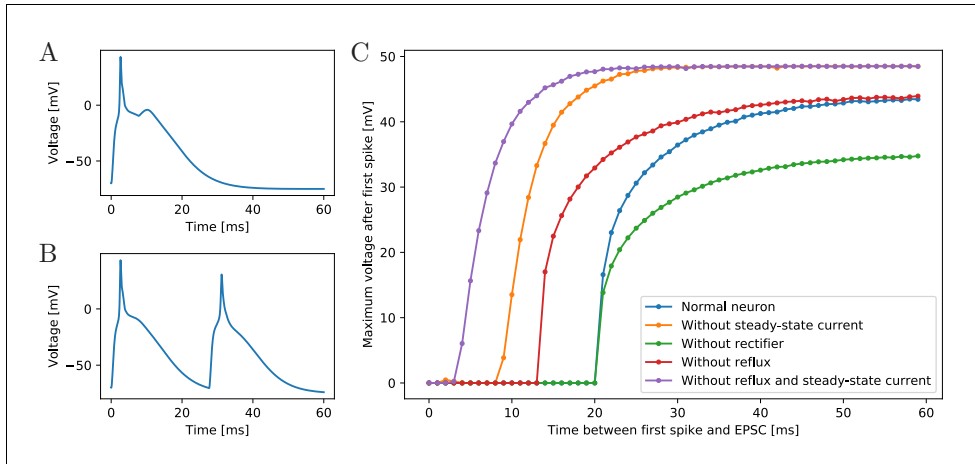

**Figure 3.** Excitability of an MNN neuron after spiking. (**A,B**) Voltage of an example neuron receiving two identical EPSCs (**A**) 7 ms apart and (**B**) 25 ms apart. (**C**) Maximum voltage reached in response to the second EPSC for different time lags between the inputs. The first EPSC always generates a spike. The abscissa displays the time differences between its peak and the onset of the second EPSC. The ordinate displays the highest voltage reached after the end of the first spike, defined as reaching 0 mV from above. A plotted value of 0 mV means that the neuron did not exceed 0 mV after its first spike.

to about 5 ms (purple trace in *Figure 3C*). In contrast, deactivation of the synaptic rectifier does not shorten the refractory period, but reduces the amplitude of the action potential, since the reversal potential of the channels is +4 mV. The synaptic rectifier thus allows spike peaks to more clearly exceed the +20 mV threshold for synaptic transmission activation. It may therefore increase the reliability of signal conduction in the MNN.

## Modeling the motor nerve net
### Qualitative dynamics
Given the described qualitative properties of its neurons and synapses, we can explain the main feature of the MNN, namely throughconductance without pathological firing: In fact, the properties of the MNN indicate that during the activation wave following an arbitrary initial stimulation of the network, every neuron spikes exactly once. Generally, this is the case in a network where (i) the synapses are bidirectional, (ii) a presynaptic action potential evokes action potentials in all non-refractory postsynaptic neurons and (iii) the refractory period is so long that there is no repetitive firing in two neuron systems.

This becomes clear if we think of the nerve net as a connected undirected graph with neuron dynamics evolving in discrete time steps. The undirectedness of the graph reflects the synaptic bidirectionality, point (i) above. We assume that it takes a neuron one time step to generate an AP; its postsynaptic neurons that are resting generate an AP in the next time step, see point (ii). After an AP, a neuron is refractory for at least one time step and thereafter becomes resting, ensuring (iii). More formally speaking, each vertex can be in one of three states in any time step: resting, firing, refractory. The state dynamics obey the following rules:

1. If a vertex is firing at time step $t_i$, every connected, resting vertex will fire at $t_{i+1}$.
2. If a vertex is firing at $t_i$, it will be refractory at $t_{i+1}$.
3. If a vertex is refractory at $t_i$, it will be resting at $t_{i+1}$.

If in such a graph a number of vertices fires at $t_0$ while the other vertices are resting (initial stimulation), every vertex will subsequently fire exactly once: Obviously any vertex $X$ will be firing at $t_x$, where $x$ is the minimum of the shortest path lengths to any of the vertices firing at $t_0$. Further, if a vertex $Y$ is firing at $t_y$, where $y = x + s$, there must be a vertex $X$ firing at time $t_x$ with a path from $X$ to $Y$ with path length $s$. We will now assume that a vertex $X$ is not only firing at $t_x$ but also at $t_{x'}$ and show that this is impossible as it leads to a contradiction: We have $x' > x$ since $t_x$ is by definition the

first time that $X$ fires after the initial stimulation. Since the vertex is refractory at $t_{x+1}$ and resting at $t_{x+2}$, even $x'>x+2$ holds. Let $x'=x+j$ where $j>2$. This implies that at $t_x$ a vertex $Y$ must be firing, with a path between $X$ and $Y$ of length $j$, along which the firing spreads from $Y$ towards $X$. There is, however, also a chain of firing traveling along this path from $X$ to $Y$. If $j$ is even this results in two vertices in the center of the path firing right next to each other at $t_{x+\frac{j}{2}}$. After that both vertices are refractory and no other vertex along this path is firing. If $j$ is odd there are two vertices firing at $t_{x+\frac{j-1}{2}}$ with a single vertex separating them. This vertex fires in the next time step, but since both neighboring vertices on this path are then refractory, no vertex along this path fires after that. Both cases contradict the initial assumption that $X$ spikes at $t_{x+j}$. We may thus conclude that $X$ fires only once.

## Geometry

To model the MNN in more detail, we uniformly distribute the developed Hodgkin-Huxley type neurons on a disc representing the subumbrella of a jellyfish with diameter 4 cm. Its margin and a central disc are left void to account for margin and manubrium (see Materials and methods for further details). Eight rhopalia are regularly placed at the inner edge of the margin. We model their pacemakers as neurons which we stimulate via EPSCs to simulate a pacemaker firing. The neurons are geometrically represented by their neurites, modeled as straight lines of length 5 mm (*Horridge, 1954a*). At the intersections of these lines lie connecting synapses (*Anderson, 1985*; *Anderson and Grünert, 1988*). All synapses are bidirectional and have the same strength, sufficient to evoke an AP in a postsynaptic neuron. We incorporate neurite geometry and relative position into our single compartment models by assuming that the delay between a presynaptic spike and the postsynaptic EPSP onset is given by the sum of (i) the traveling time of the AP from soma to synapse on the presynaptic side, (ii) the synaptic transmission delay and (iii) the traveling time of the EPSC from synapse to soma on the postsynaptic side. The traveling times depend linearly on the distances between synapse and somata; for simplicity, we assume that AP and EPSC propagation speeds are equal. In agreement with *Anderson (1985)*, the total delays vary between 0.5 ms and 1.5 ms.

Interestingly, the preferred spatial orientations of MNN neurites along the subumbrella are related to neuron position. *Horridge (1954a)* reports the following observations:

- Near the rhopalia, most neurites run radially with respect to the jellyfish center.
- Near the outer bell margin and between two rhopalia, most neurites follow the edge of the bell.
- Closer to the center of the subumbrella there is no obvious preferred direction.

To incorporate these observations, we draw the neurite directions from distributions whose mean and variance depend appropriately on neuron position. Specifically, we use von Mises distributions for the angle, which are a mathematically simple approximation of the wrapped normal distribution around a circle (*Mardia and Jupp, 1999*).

The neurite orientation structure may emerge due to ontogenetic factors: In the complex life cycle of scyphozoans, juvenile jellyfish start to swim actively during the ephyra stage. In this stage, the jellyfish has some visual similarity to a starfish, with a disc in the center containing the manubrium, and eight (or more) arms, one per rhopalium, extending from it. The motor nerve net is already present in the ephyra and extends into its arms (*Nakanishi et al., 2009*). As the jellyfish matures, the arms grow in width until they fuse together to form the bell. MNN neurites simply following the directions of growth would thus generate a pattern as described above: Neurites in the center disc may not have a growth direction or constraints to follow, therefore there is no preferred direction. When the ephyral arms grow out, neurites following the direction of growth run radially. Also the geometric constraints allow only for this direction. Neurons that develop in new tissue as the arms grow in width to form the bell orient circularly, following the direction of growth.

## Network statistics

There are, to our knowledge, no estimates on the number of neurons in a scyphozoan MNN; only some measurements for hydrozoans and cubozoans exist (*Bode et al., 1973*; *Garm et al., 2007*). However, *Anderson (1985)* measured the synaptic density in the MNN of *Cyanea capillata*: the average distance between two synapses along a neurite is approximately 70 µm. For a neuron of 5 mm

length, this translates to roughly 70 synapses placed along its neurites. To obtain an estimate for the number of MNN neurons from this, we generate model networks with different neuron numbers, calculate their average synaptic distances and compare them with the experimentally observed values (see *Figure 4A*). We find that in a von Mises MNN, about 8000 neurons yield the experimentally measured synaptic density, while the uniform MNN requires about 5000 neurons. In general, for a fixed number of neurons, a von Mises MNN is more sparsely connected than a uniform MNN: The biased neurite direction at the bell margin of a von Mises MNN (see Figure 18 in Materials and methods) implies that neurons in close proximity have a high probability of possessing similarly oriented neurites. This decreases their chance of overlap and thus the number of synapses.

## Waves of activation in the MNN

Our numerical simulations confirm that firing of a pacemaker initiates a wave of activation where every MNN neuron generates exactly one AP (see *Figures 5* and *6* for an illustration). The activity propagates in two branches around the bell. These cancel each other on the opposite side. During the wave, all other pacemakers fire as well, which presumably resets them in real jellyfish. In a uniform MNN the wave spreads rather uniformly (*Figure 6*). In a von Mises MNN, the signal travels fastest around the center of the jellyfish and spreads from there, sometimes traveling a little backwards before extinguishing (*Figure 5*).

*Gemmell et al. (2015)* observed a delay between the muscle contractions on the initiating and the opposite side of about 30 ms (std. dev. 14 ms), in *Aurelia aurita* of 3–4 cm diameter. This delay should directly relate to the propagation of neural activity. We thus compare it to the delay between spiking of the initiating pacemaker and the opposing one in our model MNNs. We find that both our von Mises and uniform MNNs can generate delays within one standard deviation of the measurements, see *Figure 4B*. Our simulations indicate that MNN networks typically have 4000 neurons or more, as the propagation delays obtained for jellyfish with 3 and 4 cm diameter start to clearly bracket the experimentally found average at this size.

*Figure 4B* shows that the delay decreases with neuron density. On the one hand, this is because in denser networks among the more synaptic partners of a neuron there will be some with better positions for fast wave propagation; in other words, the fastest path from the initiating pacemaker to the opposing one will be better approximated, if the neurons have more synaptic partners to which the activity propagates. On the other hand, there is a decrease of delay due to stronger stimulation of neurons in denser networks: a postsynaptic neuron fires earlier if more presynaptic neurons have fired, since their EPSCs add up.

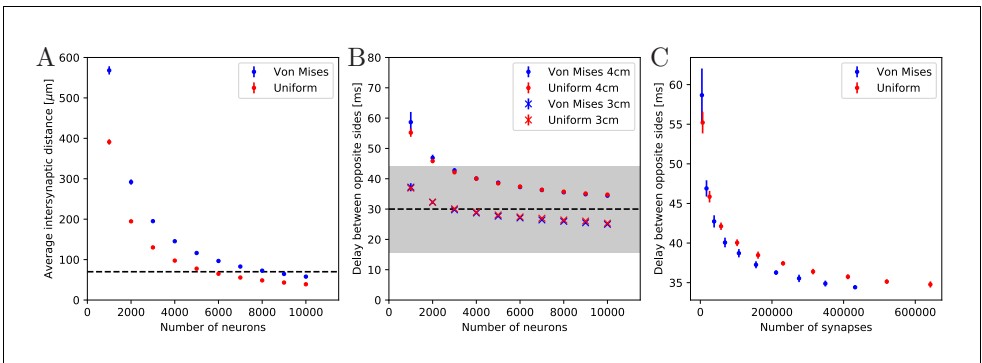

**Figure 4.** Synaptic density and activity propagation speed in von Mises and uniform MNNs. (**A**) Average intersynaptic distance as a function of neuron number in von Mises and uniform MNNs. The dashed line indicates 70 μm (*Anderson, 1985*). (**B**) Delay between the spike times of the pacemaker initiating an activation wave and the opposing one, for different MNN neuron numbers. Displayed are results for model jellyfish with 3 cm and 4 cm diameter. The dashed line indicates the experimentally measured average delay of 30 ms between muscle contractions on the initiating and the opposite side of *Aurelia aurita* (*Gemmell et al., 2015*); the gray area shows its ±1 std. dev. interval. (**C**) Delays measured in (**B**) for the 4 cm jellyfish, plotted against the average number of synapses in MNNs with identical size. Measurement points are averages over 10 MNN realizations; bars indicate one standard deviation.

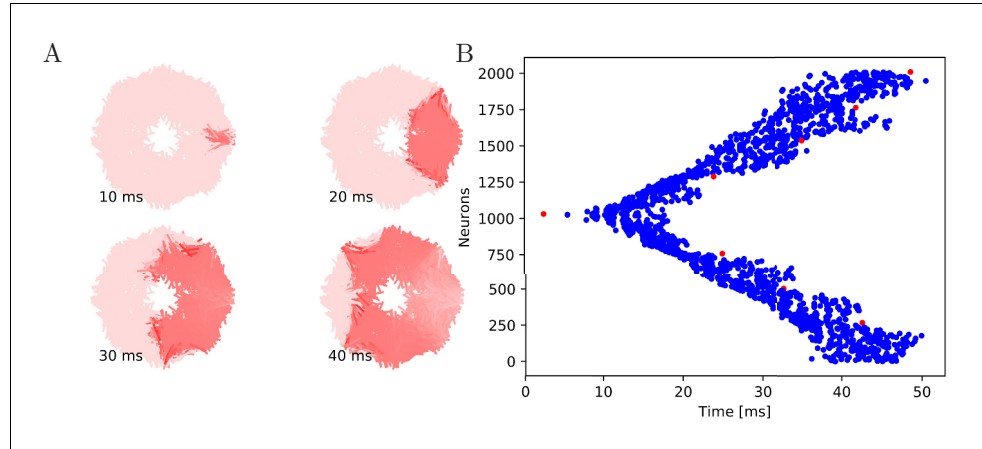

**Figure 5.** Wave of activation in a von Mises MNN with 2000 neurons. (**A**) Activity of each neuron at different times after stimulation of a single pacemaker neuron. Color intensity increases linearly with neuron voltage. (**B**) Spike times of the same network. Neurons are numbered by their position on the bell. Red dots represent the pacemakers inside one of the eight rhopalia. The neurite orientations are distributed according to location-dependent von Mises distributions.

**Figure 5—animation 1.** Example animation of *Figure 5* (A).

https://elifesciences.org/articles/50084#fig5video1

Both von Mises and uniform MNNs reach similar propagation speeds with the same number of neurons (*Figure 4B*), but von Mises MNNs have fewer synapses (*Figure 4C*). This implies that von Mises MNNs create more optimal paths of conduction. Indeed, neurons near the pacemaker preferably orient themselves radially towards the center of the subumbrella, and thus quickly direct the activity toward the opposite side. Since transmitter release consumes a significant amount of energy (*Niven, 2016*), we conclude that von Mises networks are more efficient for fast through-conduction than uniform ones.

## Cutting experiments

To further illustrate that the nerve net is through-conducting even when its structure is heavily damaged, we replicate some of the cutting experiments by *Romanes (1885)*. In these experiments, Romanes cut the umbrella of the jellyfish several times and observed that the activity is able to spread through small bottlenecks created by these cuts. To test if our MNN model reproduces this

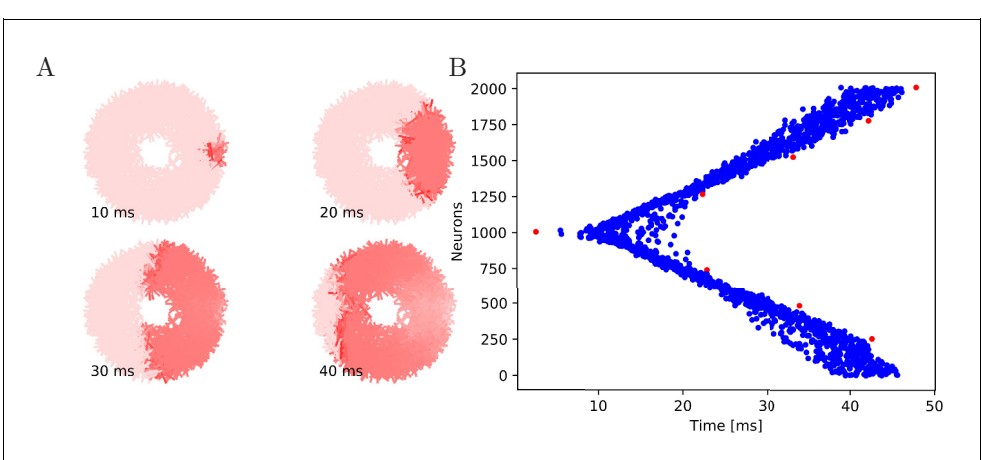

**Figure 6.** Wave of activation in a uniform MNN with 2000 neurons. Setup similar to *Figure 5*, but the neurite orientations are uniformly distributed.

behavior, we simulate cuts by straight line segments, assuming that if a neuron intersects with that line segment, the larger part (containing the soma) will survive and still transmit and receive potentials via the leftover intact synapses, while the smaller part (without the soma) dies off. In the first cutting experiment, an inner disc on the subumbrella is almost completely cut off from an outer ring. The two sections are only connected by a small patch (*Figure 7*). In the second experiment, 16 cuts are placed radially in an interdigitating fashion around the umbrella. The signal has to travel between the interleaving cuts (*Figure 8*). In both cases, we find that the excitation wave is able to travel through the whole nerve net, with von Mises or uniform neurite orientation (*Figure 9*). This again confirms our analytical result: the through-conducting property is preserved and every neuron in the network fires once, no matter how the neurons are connected.

## A model of straight swimming

### MNN activation and swimming strokes

To analyze the swimming behavior, we employ a 2D hydrodynamics simulation of a cross section of the jellyfish bell. We assume that MNN neurons synaptically connect to muscles that lie in the same region (see Materials and methods for details). APs in the neurons evoke stereotypical contractions of the muscles. These add up to large muscle forces contracting the bell. Their interaction with the elastic forces of the bell and the hydrodynamics of the media in- and outside the bell determines the dynamics of the swimming stroke. *Figure 10* shows a representative time series of such a stroke. The left hand side pacemaker initiates a wave of MNN activation, which in turn triggers a wave of contraction around the subumbrella. Because the MNN activation wave is fast compared to muscle contraction and swimming movement, the motion is highly symmetrical. As a result, the jellyfish hardly turns within a stroke.

We can qualitatively compare the simulated swimming motion to that of real jellyfish by considering the formation of vortex rings. Earlier research suggests that the formation of two vortex rings pushes oblate jellyfish, such as *Aurelia*, forward (*Dabiri et al., 2005*; *Gemmell et al., 2013*; *Gemmell et al., 2015*). In a 2D cross-section, a vortex ring is reflected by a vortex pair with opposing spin. We find indeed that two such vortex pairs are shed off near the bell margin (see *Figure 10*). The first pair is shed off during the contraction and the second one during the relaxation. The second pair slips under the jellyfish bell, which provides additional forward push (*Gemmell et al., 2013*). After the swimming stroke, the vortex rings in real jellyfish leave the bell and tend to stretch out (*Dabiri et al., 2005*). In contrast, in our 2D model, the vortex pairs move further into the bell and interact with it for a longer time. This has been observed in previous 2D models of oblate

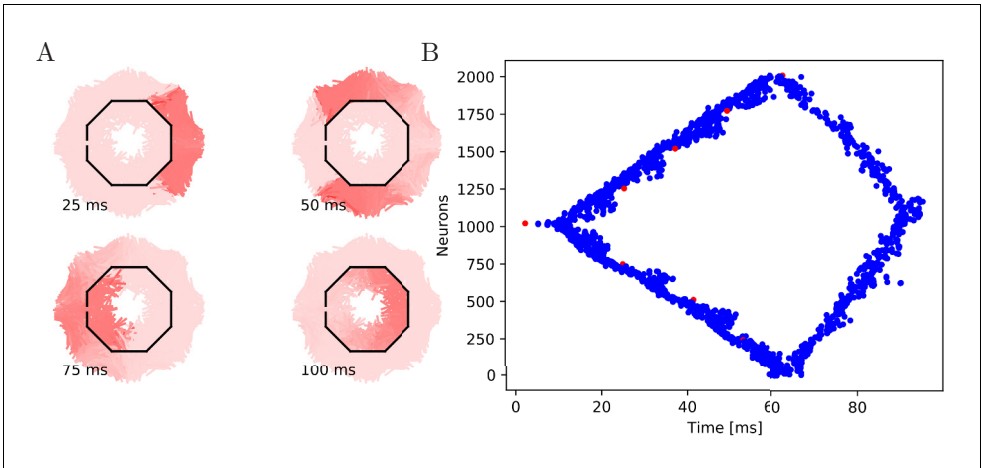

**Figure 7.** Propagation in a circularly cut von Mises MNN with 2000 neurons. Setup similar to *Figure 5*, but black line segments indicate cuts through the nervous system where neurites are severed. Cuts are placed along the outline of an octagon with a small gap through which the signal can propagate to the central neurons.

**Figure 7—animation 1.** Example animation of *Figure 7* (A).

https://elifesciences.org/articles/50084#fig7video1

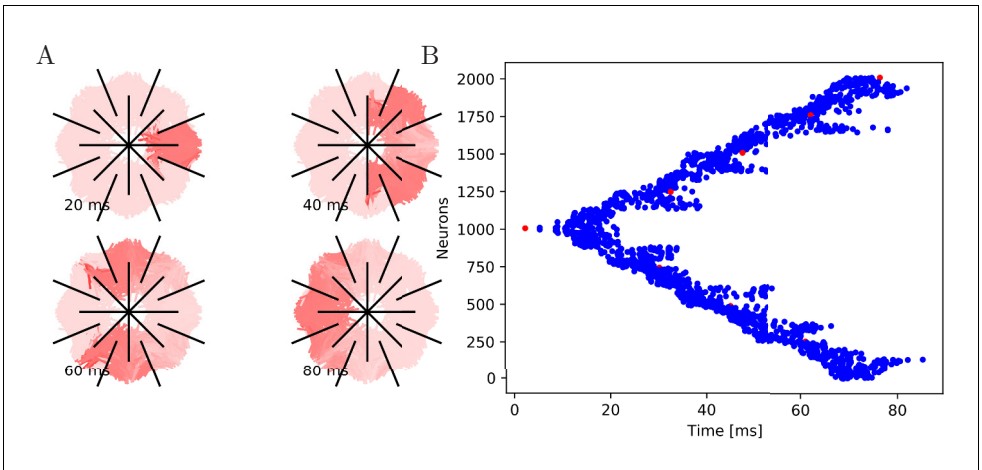

**Figure 8.** Propagation in a radially cut von Mises MNN with 2000 neurons. Setup similar to *Figure 7*, but the cuts are placed radially creating a zig-zag patterned bell.
**Figure 8—animation 1.** Example animation of *Figure 8* (A).
https://elifesciences.org/articles/50084#fig8video1

jellyfish, even with prescribed bell deformation and is likely due to the different behavior of 2D and 3D vortices (*Herschlag and Miller, 2011*). Simulations of more prolate jellyfish show less discrepancy.

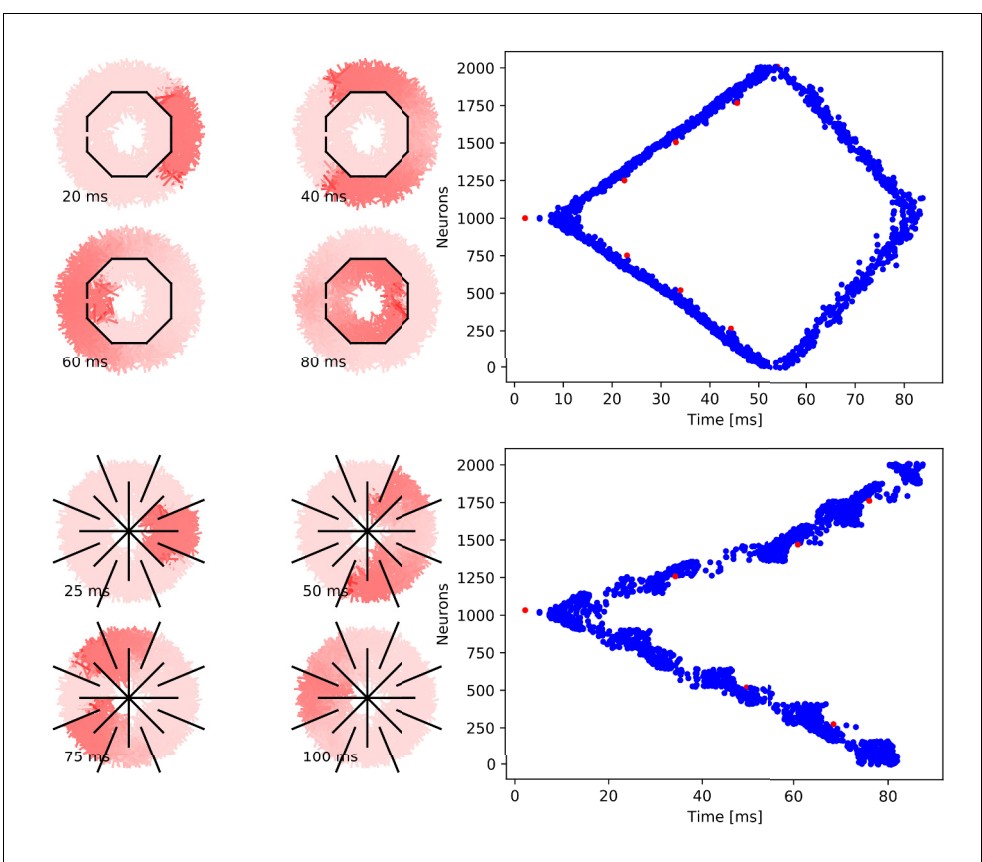

**Figure 9.** Cutting experiments in uniform MNNs with 2000 neurons. Setup similar to *Figures 7* and *8*, but the neurite orientations are uniformly distributed.

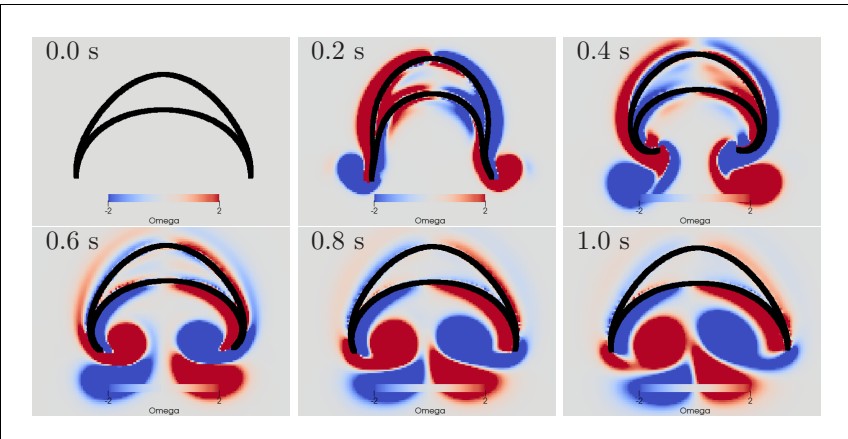

**Figure 10.** Swimming stroke evoked by a wave of activation in the MNN. The panels show the dynamics of the bell surface (black) and internal and surrounding media (grey), in steps of 200 ms. Coloring indicates medium vorticity $\Omega$ (in $1/s$), blue a clockwise eddy and red an anticlockwise one. In this and all following figures, it is the pacemaker on the left hand side of the bell that initiates MNN activation. Further, if not stated otherwise, the MNN has 10,000 neurons.

**Figure 10—animation 1.** Animation of the swimming motion.

https://elifesciences.org/articles/50084#fig10video1

**Figure 10—animation 2.** Animation of both a swimming stroke and the corresponding MNN activation as seen in *Figure 5*.

https://elifesciences.org/articles/50084#fig10video2

---

*McHenry and Jed (2003)* measured changes in the bell geometry of *Aurelia aurita* during its swimming motion. When tracking the same data in our simulations for our standard parameters, we find qualitatively similar time series (see *Figure 11* blue). In particular, the sequence of changes in the bell geometry agrees with that of real jellyfish (*Figure 11A,B*). During the contraction phase, the bell diameter shrinks and the bell height increases. The bell margin begins to bend outward as the jellyfish contracts and folds inward during the relaxation of the bell. The margins of the real jellyfish bend less than those of our model jellyfish (*Figure 11C*). Their higher stiffness may originate from passive resistance of the probably inactive radial muscles. The speed profile in the experiments shows broader peaks and a longer continuation of forward movement after bell relaxation compared to our (*Figure 11D*) and previous 2D models (*Herschlag and Miller, 2011*). In particular, the models produce negligible forward momentum during the relaxation phase in oblate jellyfish. This may again be due to differences in vortex dynamics in 2D and 3D, as a 3D model does not show this discrepancy (*Park et al., 2014*). To test if the quantitative agreement of our model with the measurements can be improved, we adjusted the bell size and spring parameters (*Figure 11* orange). While this leads to a better agreement of the margin bending, the speed profile does not improve, unless we switch to a more prolate bell shape (not shown). This supports the idea that a 2D model of oblate jellyfish is unable to reproduce the real rowing mechanism.

## Influence of network size

To quantify the effects of MNN size on swimming, we evaluate travel distances and changes in orientation, see *Figure 12*. We find that the typical total distance traveled by individual jellyfish increases with network size (*Figure 12A,B*), while the variance and thus the typical distance traveled sideways and the typical angular movement decrease (*Figure 12A,C A,D*). This can be explained by the higher temporal and spatial coherence in the activation waves of larger MNNs. They arise from larger throughconductance speed, see *Figure 4*, and from more uniform neuron density and muscle innervation: Since neurons are distributed uniformly in space, the fluctuations of local neuron density relative to its mean decreases with increasing neuron number. This implies that the relative fluctuation in the number of neurons innervating the different muscle segments decreases. With small MNNs, random fluctuations in the number of innervating neurons are likely to lead to a spatial

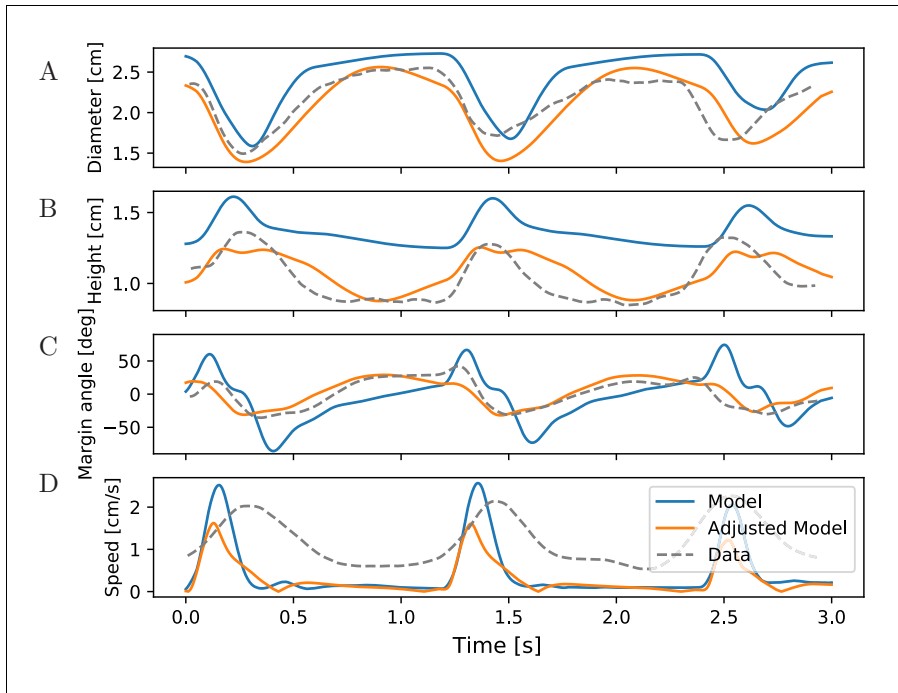

**Figure 11.** Characteristics of bell shape during swimming. Dynamics of (**A**) bell diameter, (**B**) bell height and (**C**) the orientation of the margin of the bell relative to the orientation of the bell as a whole, during a sequence of swimming strokes as in *Figure 10A*, initialized in intervals of 1.2 s. (**D**) Corresponding speed profile. Shown are models with our standard parameters (blue) and manually adjusted parameters (orange) to match the experimentally found traces (gray) in *McHenry and Jed (2003)* (Fig. 2 ibid., adapted with permission from Journal of Experimental Biology).

imbalance of contraction force that is sufficient to generate marked sideways movement and turning. Generally, the variance of a characteristic sampled over different MNN realizations decreases as the number of neurons increases, because the decrease of relative local density fluctuations implies that the network ensembles become more homogeneous.

## A model of turning

### The mechanism of turning

Finally, we investigate whether the contraction of the bell margin due to DNN activity can lead to a turning mechanism similar to the one suggested by *Gemmell et al. (2015)*. This study observed that the margin at the inside of a turn was stiffened, which may explain the weaker vortex and thrust generation there and the resulting turn around it. The DNN was suggested to control the stiffening via radial muscles. To test this mechanism, we augment our jellyfish model by a DNN similar to the MNN (see *Figure 1*). Its neurons are governed by the same equations, but the neurites are only 2 mm long (*Passano and Passano, 1971*) and we assume for simplicity that their orientation is unbiased. The DNN extends 0.25 cm further than the MNN into the bell margin, where the radial muscles are situated. The DNN controls the activity of the radial muscles in the same manner as the MNN controls the activity of the circular ones. Similarly to the MNN a wave of DNN activity is initiated in the rhopalia.

We find that a simultaneous activation of the DNN and the MNN indeed leads to a turn, see *Figure 13*. The jellyfish turns towards the origin of the contraction wave if both MNN and DNN are stimulated at the same time. The radial muscles of the bell margin on the stimulated side contract simultaneously with the circular muscles such that the bell margin stiffens up and does not bend outwards during the contraction of the bell, cf. the left hand side margin in *Figure 13*. Because the water resistance is increased on this side, the contraction is slowed down. Due to the different conduction speeds of MNN and DNN, the circular muscles on the other side contract before the radial

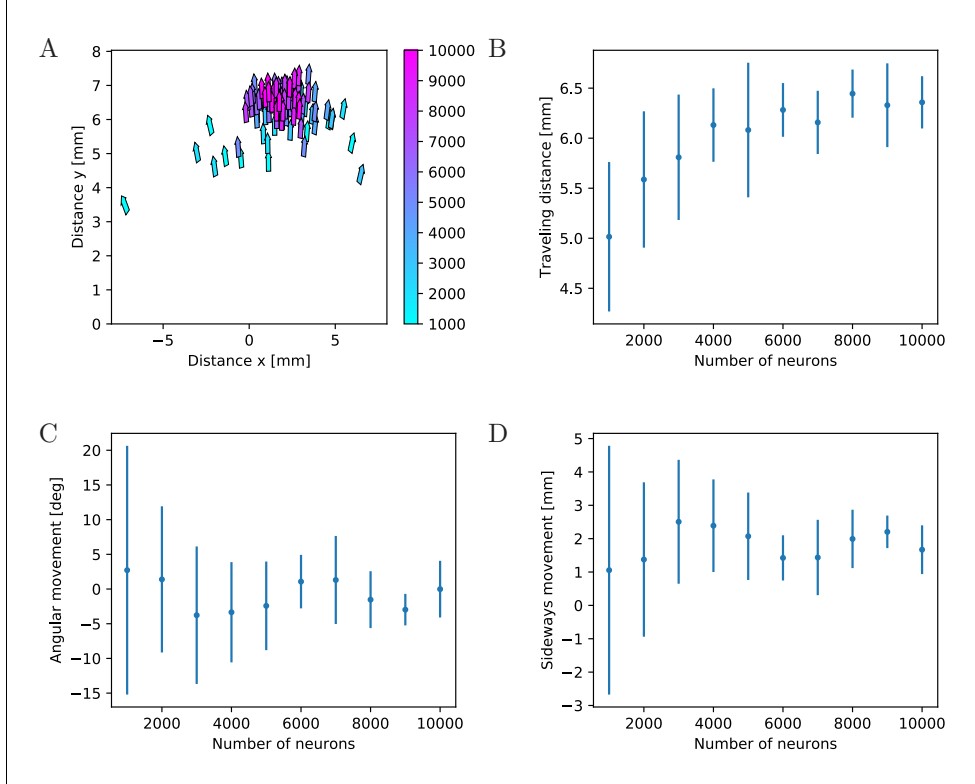

**Figure 12.** Characteristics of swimming strokes for different MNN sizes. **A** shows the distance traveled within a single swimming stroke (origins of arrows) and the orientation after the stroke (direction of arrows) for 100 jellyfish with different MNNs. Color indicates the MNN sizes, which range in 10 steps from 1000 to 10,000. (**B, C, D**) visualize the dependence of the distributions of swimming characteristics on MNN size. **B** shows the total distances traveled, **C** the angular movements (i.e. angular changes in spatial orientation, in degrees) and **D** the distances moved perpendicularly to the original orientation of the jellyfish. Measurement points are the averages of the 10 jellyfish with MNNs of the same size in **A**, bars indicate one standard deviation.

muscles. The stroke is therefore similar to that during straight swimming, leads to a stronger contraction and turns the jellyfish toward the origin of the activation wave.

The displayed dynamics are similar to those experimentally observed in *Aurelia* by *Gemmell et al. (2015)*. In particular, the jellyfish turns toward the side of initial contraction and the bell margin on the inside of the turn is contracted while the opposing one extends outwards. The margin bending in our model appears stronger than in *Gemmell et al. (2015)*. Further, the delay between the onsets of contraction on the initiating and the opposing sides is shorter in our model. Such dissimilarities may be brought into agreement by more detailed DNN and bell modeling in 3D hydrodynamic environments.

## Relative timing of MNN and DNN activation

*Passano (1965)* and *Passano (1973)* found that after externally stimulating the DNN, the MNN becomes active after a significant delay. We therefore study the impact of different delays between DNN and MNN activation on the turning behavior, see *Figure 14*. For small delays, the jellyfish turns toward the origin of the stimulation, like for zero delay (*Figure 13*) and as observed by *Horridge (1956)* and *Gemmell et al. (2015)*. As the delay increases, the jellyfish turns less. At a certain delay the turning direction changes, and the jellyfish turns more and more into the opposite direction. For even larger delays, the jellyfish again turns less and there is eventually another change of direction. The points of first direction change and maximum opposite turning depend on the speed of the DNN signal (*Figure 14*).

The first change of turning direction occurs because for sufficiently large delay between DNN and MNN the radial muscles on the side of wave initiation are already relaxing when the circular muscles

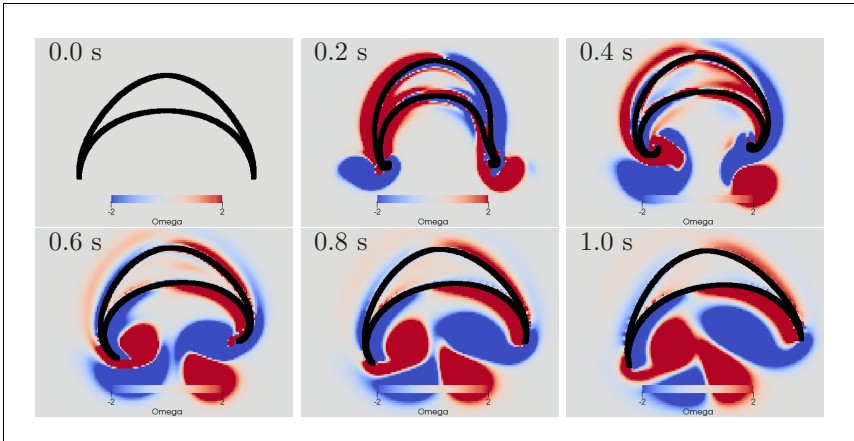

**Figure 13.** Swimming stroke evoked by simultaneously initiated waves in MNN and DNN. The activity in the DNN and MNN leads to a simultaneous contraction of the left bell margin and the left bell swim musculature near the margin. The jellyfish therefore turns in the direction of the initiating rhopalium. The DNN has 4000 neurons. MNN and further description are as in *Figure 10*.

**Figure 13—animation 1.** Animation of the swimming motion.
https://elifesciences.org/articles/50084#fig13video1

---

contract, see *Figure 15*. On the opposing side, the activity of the radial muscles then coincides with the contraction of the circular muscles. Therefore, the same mechanism that causes the turn towards the initiating rhopalium for simultaneous DNN and MNN activation lets the jellyfish now turn to the other side. This occurs although both DNN and MNN are activated by the same rhopalium. The most negative angular movement occurs at a delay that is about the conduction delay of the MNN shorter than the time it takes the DNN to conduct a signal around the bell; compare the delays at minima in *Figure 14* with the corresponding DNN conduction delays in *Figure 14* minus the MNN conduction delay of 35 ms. With such a delay, the two signals will simultaneously reach the opposing side of the bell.

This previously undescribed mechanism may explain how a jellyfish is able to avoid undesired stimuli. After it is, for example, mechanically stimulated somewhere on its bell, the corresponding

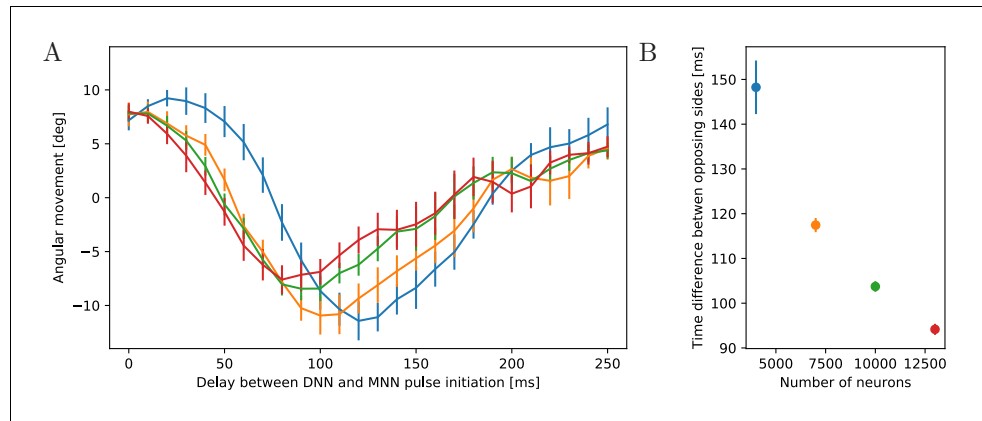

**Figure 14.** Dependence of turning on the delay between DNN and MNN activation. (A) Angular movement of model jellyfish versus delay between DNN and MNN activation. The panel displays the angular movement one second after the initiation of the MNN. Turns toward the initiating rhopalium have positive angular movements, while turns away have negative ones. Blue, orange, green and red coloring indicates DNN sizes of 4000, 7000, 10,000 and 13,000 neurons. (B) Delay between initiation of DNN activity and its reaching of the opposing side, as a function of the number of DNN neurons (similar to *Figure 4B*). Measurement points are averages over 10 realizations of MNNs with 10,000 neurons and DNNs with the indicated size, bars indicate one standard deviation.

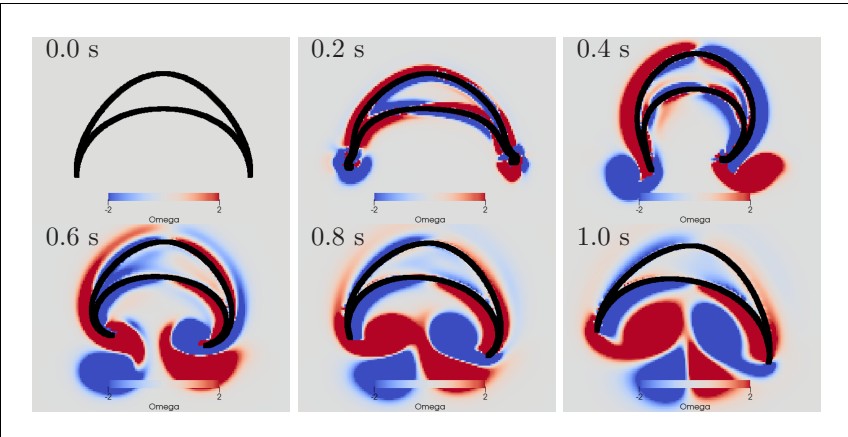

**Figure 15.** Swimming stroke evoked by sequentially initiated waves in MNN and DNN. Initiation of the MNN 120 ms after the DNN leads to a simultaneous contraction of the right bell margin and the right bell swim musculature near the margin. The jellyfish therefore turns away from the direction of the initiating rhopalium. MNN and DNN as in *Figure 13*.

**Figure 15—animation 1.** Animation of the swimming motion.
https://elifesciences.org/articles/50084#fig15video1

DNN excitation spreads and reaches the rhopalium closest to the origin of the stimulus. If the MNN would then fire immediately, the jellyfish would turn towards the stimulus. Our simulation together with the experiments by *Passano (1965)* and *Passano (1973)* let us hypothesize that the pacemaker at the rhopalia may rather fire after an appropriate delay, generated by a yet unknown mechanism. This would allow the jellyfish to flee if necessary.

## Discussion

We have built a multiscale model of the neuromuscular system of scyphozoan jellyfish on the basis of biophysical, physiological and anatomical data. Our model reproduces known experimental findings and predicts new ones across multiple scales, from ion channel dynamics over neuron and neuronal network activity to animal behavior.

We propose a Hodgkin-Huxley-type neuron model for scyphozoan MNN neurons, on the basis of voltage-clamp data (*Anderson, 1989*). The model yields an explanation for experimental findings, such as the long refractory period of MNN neurons (*Anderson and Schwab, 1983*), in terms of ion channel and synapse dynamics. Furthermore, it makes experimentally testable predictions on the time course of different ion channel activations during an AP and the effect of their blocking. The number of parameters in the model could be reduced. For example, the slow outward current does not contribute to the neuron dynamics in the considered physiological regime. It will be interesting to explore which parameter values are crucial for its functioning in the future.

We develop the idea of synaptic transmitter reflux as a natural consequence of the bidirectional synapses connecting MNN neurons (*Anderson, 1985*). Our model indicates that the synaptic reflux generates a peculiarity of the scyphozoan AP shape, namely a delayed decay or small voltage bump immediately after the return from peak AP depolarization, which is visible in experimental data (*Anderson, 1985*). Later voltage bumps occur since postsynaptic APs evoke EPSCs in the presynaptic neuron (*Figure 3A,B*; *Anderson, 1985*; *Anderson and Schwab, 1983*).

A simple, phenomenological network model qualitatively incorporating key features of MNN neurons shows why MNN and DNN do not generate pathological activity, but a single wave of activation after an initial stimulation. The model predicts that during such a wave every neuron in the nerve net fires exactly once, no matter where the initial excitation originates.

We build a biologically more detailed neuronal network model of the scyphozoan MNN by placing the developed Hodgkin-Huxley-type neurons on a 2D geometry representing the subumbrella. Based on anatomical observations (*Horridge, 1954a*), we propose that their neurite orientations are

distributed according to location-dependent von Mises distributions. We study the dynamics of these von Mises MNNs and compare them to MNNs with uniformly distributed neurite orientations. Similarly, we build a model for the DNN. Since electrophysiological data on the DNN is so far missing, we use the same neuron model as for the MNN, except for shorter neurites (*Passano and Passano, 1971*). For simplicity, we draw the DNN neurite orientations from a uniform distribution. The real networks are more complex. In particular, the DNN extends into the exumbrella, the manubrium and the tentacles *Horridge (1956)*; its neurites possess a bias towards a radial orientation (*Satterlie and Eichinger, 2014*). Furthermore, immunohistochemical staining suggests that the MNN innervates the bell margin, where the neurons form a 'pseudo-nerve ring' (*Satterlie and Eichinger, 2014*), which may mediate the interaction with the tentacles. While at least some of these complexities are certainly important for the behavior and survival of *Aurelia*, we expect them to be less relevant for its swimming dynamics as depicted in the present study. As an example, the radial orientation preference of the DNN neurites lowers the speed of activation spread (*Figure 16*). This does not change the qualitative turning behavior and may have a smaller quantitative impact than the (unknown) number and dynamics of individual DNN neurons.

Both our von Mises and uniform MNNs can reproduce the experimentally observed through-conduction delay of MNN activation waves. Von Mises MNNs are, however, more cost efficient in the sense that their waves require fewer synaptic transmitter releases to reach the same delay. The experimentally found biological features of the network structure thus provide a partial optimization compared to homogeneous random networks. We suggest that the structure may emerge in a simple manner as the neurites follow the directions of growth and geometric constraints during ontogenesis.

Our model suggests two estimates of the unknown number of neurons in a scyphozoan MNN. The first one is purely geometrical, based on our network structure and the average distance of synapses on neurites measured by *Anderson (1985)*. The second one accounts for the network dynamics and compares throughconduction delays in our models with experimentally measured ones in *Gemmell et al. (2015)*. The estimates indicate that the number of neurons is of the order of 10,000 neurons in jellyfish of about 4 cm diameter. Possible error sources of the estimates include the mixing of data from animals of different species (*Cyanea capillata* in *Anderson, 1985*; *Anderson, 1989* and *Aurelia aurita* in *Horridge, 1954a*) and sizes, distributed neurite lengths and the presence of multipolar cells and multiple synapses between neurons (*Horridge, 1954a*; *Anderson, 1985*). The obtained neuron numbers are within the range found for other cnidarians: hydrozoans and cubozoans have approximately 5000 to 20,000 neurons (*Bode et al., 1973*; *David, 1973*; *Garm et al., 2007*). Our turning experiments imply that larger angular displacements occur for smaller and therefore slower DNNs (see *Figure 14*). This suggests that the neuron density of the DNN in real jellyfish is small. Immunohistochemical staining experiments indeed find that the neuron density of the DNN is lower that that of the MNN (*Satterlie and Eichinger, 2014*). The above-mentioned radial orientation preference may serve to further slow the activity spread in the DNN down (*Figure 16*).

To connect neural activity to behavior, we develop a model for the muscle system and the elastic bell of *Aurelia aurita*. The MNN evokes the contractions of the swim musculature. We place

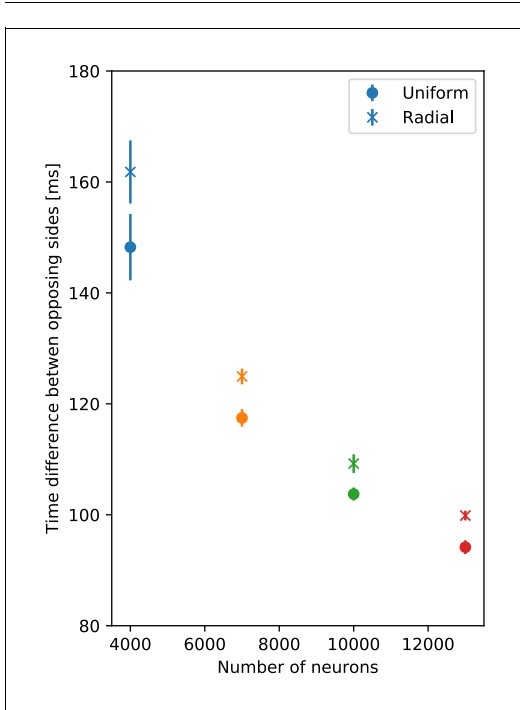

**Figure 16.** Comparison of propagation speed in DNNs. Delay between initiation of DNN activity and its reaching of the opposing side, as a function of the number of DNN neurons (similar to *Figure 14*). Neurite orientation in these nerve nets is either uniform or has a radial bias.

the resulting model jellyfish in a hydrodynamic environment and simulate its swimming behavior. To reduce the duration and complexity of the hydrodynamics simulations, we consider a 2D jellyfish model and environment. We observe shedding of vortex pairs in the surrounding medium and, after appropriately adjusting parameters of the fluid-structure simulation, bell geometry dynamics similar to experimental observations (*McHenry and Jed, 2003*; *Dabiri et al., 2005*; *Gemmell et al., 2013*). The restriction to a 2D simulation setup entails limitations, at least for obtaining quantitatively accurate results: In 2D vortex pairs can move independently from one another, while 3D vortex rings move as one unit during real jellyfish swimming (*Dabiri et al., 2005*). Further, vortex rings in 3D expand while the corresponding vortex pairs with opposite vorticity in 2D approach each other. As a result, in our simulation the vortex pair released during the relaxation moves further into the jellyfish bell than a real vortex ring would. The difference in vortex dynamics may explain that our model jellyfish stops moving forward quickly during relaxation after a stroke in contrast to data (cf. *McHenry and Jed, 2003*; *Gemmell et al., 2013*). Other researchers found similar limitations when simulating oblate jellyfish in 2D (*Herschlag and Miller, 2011*). Previous work in 2D has only looked at a symmetric swimming motion, where vortex pairs are shed off perfectly symmetrically (*Rudolf and Mould, 2010*; *Herschlag and Miller, 2011*; *Gemmell et al., 2013*; *Hoover and Miller, 2015*). However, in our simulations, the contractions are slightly asymmetric, due to the throughconduction delay in the MNN. Since in 2D the resulting vortices move under the jellyfish bell and stay and accumulate there, they exert a strong asymmetric force after several swimming strokes initiated at the same rhopalium. To counteract this effect, we slightly increase the viscosity of the surrounding medium. A simple model of a contraction wave with finite propagation speed has been tested in a 3D jellyfish simulation in *Hoover (2015)*. They found that turning reduces with increasing propagation speed. We observe this as well when increasing the number of neurons in the MNN, which increases the propagation speed of its activation and the induced muscle contraction wave. A larger MNN also increases the distance traveled after each stroke, enhancing the swimming speed.

We find that the details of the muscle dynamics are not crucial for the effective swimming motion and that our model produces a swimming motion that appears realistic for a wide variety of parameters, with the restrictions discussed above.

Based on experimental findings (*Gemmell et al., 2015*), we incorporate radial muscles in the margin of our jellyfish model. They are activated by the DNN. If the DNN and the MNN are initiated at the same time by a rhopalium, they evoke a simultaneous contraction of the nearby bell margin and radial swim muscles. Similar to the experimental observations, we find that this turns the jellyfish towards the initiation site (*Gemmell et al., 2015*). Such voluntary turning is large compared to involuntary turning during straight swimming strokes for the estimated number of MNN neurons.

After mechanical stimulation the DNN generates a wave of activation, which in turn initiates an MNN wave at the closest rhopalium (*Horridge, 1956*). A turn toward this rhopalium and thus toward the site of stimulation may often be undesired. Our simulations indicate that appropriate delays between MNN and DNN activation induce turns away from the stimulation site. Strongest such turns occur for delays that let both excitation waves reach the opposite side at the same time. We hypothesize that the rhopalia generate appropriate delays and allow the jellyfish to avoid predators or crashing into obstacles (*Albert, 2008*). This previously unknown level of control may be experimentally detected by measuring the timing of DNN and MNN activity, similar to *Passano (1965)*, while simultaneously recording the swimming motion of the jellyfish.

In our current model, the MNN and DNN are stimulated by an artificially induced spike in one of their neurons at the location of a rhopalium. For a more complete modeling of the nervous system, future research should develop a model for pacemakers and their activity. This requires further experiments on their response properties and sensory information integration (*Nakanishi et al., 2009*; *Garm et al., 2006*). Such data will also be key to test our prediction of the jellyfish's ability to avoid predators or obstacles by turning away from them. This ability might, in addition to different timings of DNN and MNN activity, use some form of multisensory integration differentiating threats from harmless stimuli.

To conclude, in this study, we built the first comprehensive model of the neuromuscular system of a cnidarian. Specifically, we considered the jellyfish *Aurelia aurita*. This is particularly relevant due to the position of jellyfish in the evolutionary tree and their highly efficient swimming motion. Our model reproduces experimental data on multiple scales and makes several experimentally testable predictions. The simulations suggest that the simple nerve net structure may be optimized to

**Table 1.** Neuron model parameters.

| Variable | Value | Unit |
|---|---|---|
| $C_m$ | 1 | pF |
| $g_I$ | 345 | nS |
| $g_{FT}$ | 39.8 | nS |
| $g_{ST}$ | 27.2 | nS |
| $g_{SS}$ | 10.8 | nS |
| $g_L$ | 953 | pS |
| $E_I$ | 76.7 | mV |
| $E_O$ | -84.6 | mV |
| $E_L$ | -70 | mV |
| $p_a$ | 1.77 | |
| $p_b$ | 4.82 | |
| $p_c$ | 8.64 | |
| $p_d$ | 2.51 | |
| $p_e$ | 3.85 | |
| $p_f$ | 1.15 | |
| $p_g$ | 1 | |
| $V_{1/2_a}$ | -2.02 | mV |
| $V_{1/2_b}$ | -10.94 | mV |
| $V_{1/2_c}$ | 2.4 | mV |
| $V_{1/2_d}$ | $2.21 \cdot 10^{-2}$ | mV |
| $V_{1/2_e}$ | 10.65 | mV |
| $V_{1/2_f}$ | -10.01 | mV |
| $V_{1/2_g}$ | 48.58 | mV |
| $\rho_a$ | 3.99 | mV |
| $\rho_b$ | -13.03 | mV |
| $\rho_c$ | 22.55 | mV |
| $\rho_d$ | -8.97 | mV |
| $\rho_e$ | 26.43 | mV |
| $\rho_f$ | -4.57 | mV |
| $\rho_g$ | 22.41 | mV |
| $C_{\text{base}_a}$ | $5.2 \cdot 10^{-1}$ | ms |
| $C_{\text{base}_b}$ | 1.3 | ms |
| $C_{\text{base}_c}$ | $1.65 \cdot 10^{-1}$ | ms |
| $C_{\text{base}_d}$ | 2.73 | ms |
| $C_{\text{base}_e}$ | 1.13 | ms |
| $C_{\text{base}_f}$ | 7.66 | ms |
| $C_{\text{base}_g}$ | 10.43 | ms |
| $C_{\text{amp}_a}$ | $4.66 \cdot 10^{-1}$ | ms |
| $C_{\text{amp}_b}$ | $2.42 \cdot 10^{-1}$ | ms |
| $C_{\text{amp}_c}$ | 7.51 | ms |
| $C_{\text{amp}_d}$ | 10 | ms |
| $C_{\text{amp}_e}$ | 16.64 | ms |
| $C_{\text{amp}_f}$ | 2 | ms |

*Table 1 continued on next page*

*Table 1 continued*

| Variable | Value | Unit |
|---|---|---|
| $C_{\mathrm{amp}_g}$ | 4.96 | ms |
| $V_{\mathrm{max}_a}$ | $-5.87 \cdot 10^{-1}$ | mV |
| $V_{\mathrm{max}_b}$ | $2.68 \cdot 10^{-1}$ | mV |
| $V_{\mathrm{max}_c}$ | -35.22 | mV |
| $V_{\mathrm{max}_d}$ | -29.96 | mV |
| $V_{\mathrm{max}_e}$ | -12.71 | mV |
| $V_{\mathrm{max}_f}$ | -34 | mV |
| $V_{\mathrm{max}_g}$ | -39.93 | mV |
| $\sigma_a$ | 1 | mV |
| $\sigma_b$ | 6.62 | mV |
| $\sigma_c$ | 23.12 | mV |
| $\sigma_d$ | 15.13 | mV |
| $\sigma_e$ | 43.6 | mV |
| $\sigma_f$ | 20 | mV |
| $\sigma_g$ | 29.88 | mV |

conduct signals across the bell. In addition, we find that the nerve nets enable a higher level of turning control than previously thought to be present in a radially symmetric organism that only receives decentralized sensory information. Our study bridges the gap between single neuron activity and behavior in a comparatively simple model organism. It lays the foundation for a complete model of neural control in jellyfish and related species and indicates that such modeling approaches are feasible and fruitful. Our bottom-up modeling methods and our results can also be useful for modeling studies of ctenophores and cnidarians like *Hydra vulgaris*, where observing the complete nervous system of a living animal is possible (*Dupre and Yuste, 2017*; *Szymanski and Yuste, 2019*). A comparative computational analysis of their different nervous system dynamics and behavior could then shed light on the early evolution of nervous systems.

## Materials and methods

### Neuron model

We use the voltage-clamp and action potential data of *Anderson (1989)* and *Anderson (1985)* to develop a biophysical single compartment model of a scyphozoan neuron. The model describes the dynamics of the neuron's membrane potential $V$ and its transmembrane currents. Following *Anderson (1989)*, we incorporate a transient inward current ($I_I$) and three outward currents: a steady-state outward current ($I_{SS}$) and a slow and a fast transient outward current ($I_{ST}$ and $I_{FT}$, respectively). Furthermore, we include a passive leak current ($I_L$). The membrane voltage thus follows the ordinary differential equation

$$C_m \frac{\mathrm{d}V}{\mathrm{d}t} = I_{\mathrm{syn}} - I_I - I_{\mathrm{FT}} - I_{\mathrm{ST}} - I_{\mathrm{SS}} - I_L, \tag{1}$$

where $C_m$ is the membrane capacitance and $I_{\mathrm{syn}}$ the synaptic input current (see next section). The currents are modeled with a Hodgkin-Huxley type gate model (*Izhikevich, 2007*). The steady-state current has a single gating variable $G_g$; exponentiation with a suitable exponent $p_g$ yields the probability that an individual channel is open. Transient currents have two gating variables, one for activation and one for inactivation. For these currents, the probability that an individual channel is open is given by the product of the two gating variables after exponentiation with suitable exponents. The transmembrane currents are thus given by

$$I_1 = g_1 G_a^{p_a} G_b^{p_b} (V - E_1),\tag{2a}$$

$$I_{FT} = g_{FT} G_c^{p_c} G_d^{p_d} (V - E_O),\tag{2b}$$

$$I_{ST} = g_{ST} G_e^{p_e} G_f^{p_f} (V - E_O),\tag{2c}$$

$$I_{SS} = g_{SS} G_g^{p_g} (V - E_O),\tag{2d}$$

$$I_L = g_L (V - E_L),\tag{2e}$$

where $g_i$, $i \in \{\mathrm{I,FT,ST,SS,L}\}$, are the peak conductances, $E_j$, $j \in \{\mathrm{I,O,L}\}$, are the reversal potentials of the currents, $G_k$, $k \in \{a,b,c,d,e,f,g\}$, are the gating variables and $p_k$ are their exponents. As suggested by *Anderson (1989)*, we assume that the three outward currents have the same reversal potential. The dynamics of a gating variable $G_k$ follow

$$\frac{\mathrm{d}G_k}{\mathrm{d}t} = (G_{k\infty} - G_k)/\tau_{G_k}.\tag{3}$$

The voltage dependence of its steady-state value $G_{k\infty}$ is given by a logistic function with slope-factor $\rho_k$ and half-maximal voltage $V_{1/2_k}$,

$$G_{k\infty}(V) = \frac{1}{1 + \exp((V_{1/2_k} - V)/\rho_k)},\tag{4}$$

and the voltage dependence of its time constant $\tau_{G_k}$ is given by a Gaussian,

$$\tau_{G_k}(V) = C_{\mathrm{base}_k} + C_{\mathrm{amp}_k} \exp\left(\frac{-(V_{\mathrm{max}_k} - V)^2}{\sigma_k^2}\right).\tag{5}$$

Here, $C_{\mathrm{base}_k}$ is the base value of $\tau_{G_k}$, $C_{\mathrm{amp}_k}$ specifies its maximum at $V = V_{\mathrm{max}_k}$ and $\sigma_k$ is the width of the Gaussian.

To fit the models for the transmembrane currents (*Equation (2)*), we extract data points from the voltage clamp experiments of *Anderson (1989)*, Fig. 5 in Ch. 19, using WebPlotDigitizer (*Rohatgi, 2019*). We simultaneously fit all 57 parameters using the L-BFGS algorithm (*Zhu et al., 1997*) to minimize the least-squared error between model and data. We apply the basin hopping algorithm (*Olson et al., 2012*) to avoid getting caught in local minima. After obtaining the parameters for the transmembrane currents, we choose the membrane capacitance $C_m$ such that an action potential has similar features as reported in *Anderson (1985)*. Concretely, we set $C_m = 1$ pF to ensure that (i) the inflection point of an action potential is close to 0 mV and (ii) it takes about 2.5 ms for an EPSP to generate an action potential, with the synaptic parameters detailed in the next section. This fits well with the capacity of a deaxonized spherical soma of diameter 5-10 µm (*Anderson, 1985*) and a specific capacitance of 1 µF/cm² (*Gentet et al., 2000*). The used model parameters can be found in *Table 1*.

## Synapse model

*Anderson (1985)* found a voltage threshold of approximately +20 mV for synaptic transmitter release in a scyphozoan synapse. In our network model, we thus assume that when a neuron reaches this threshold from below (which happens during action potentials), excitatory postsynaptic currents are evoked in the postsynaptic neurons, after a synaptic delay. The model EPSCs (*Gerstner et al., 2014*) rise with time constant $\tau_{\mathrm{rise}}$, decay initially fast with time constant $\tau_{\mathrm{fast}}$ and then tail off with a larger time constant $\tau_{\mathrm{slow}}$,

$$I_{\mathrm{EPSC}}(t) = g_{\mathrm{syn}}\left[1 - e^{-t/\tau_{\mathrm{rise}}}\right]\left[ae^{-t/\tau_{\mathrm{fast}}} + (1-a)e^{-t/\tau_{\mathrm{slow}}}\right]\Theta(t)\max\left[(E_{syn} - V), 0\right].\tag{6}$$

Here, $E_{syn}$ is the current's reversal potential, $a$ the fraction of fast decay and $\Theta(t)$ the Heaviside

theta function. The maximum function implements a synaptic rectification reported by *Anderson (1985)*: at potentials above the reversal potential synaptic currents do not reverse but stay zero. The sum of individual EPSCs evoked in a postsynaptic neuron at times $t_0, t_1 \ldots, t_n$ yields the total synaptic current $I_{\mathrm{syn}}$ entering *Equation (1)*,

$$I_{\mathrm{syn}}(t) = \sum_{i=0}^{n} I_{\mathrm{EPSC}}(t - t_i). \tag{7}$$

Model parameters can be found in *Table 2*.

## Motor nerve net

To capture the spatial properties of the nerve nets, we model the spatial geometry of MNN neurons as line segments of length 5 mm and assume that the soma is in their center (see *Figure 17*). Two neurons are synaptically connected if their neurites overlap. The transmission delay between them is given by the constant synaptic delay of 0.5 ms and the distances between the somata and the intersection $x$ of the line segments (in cm). The total delay $\rho$ of two neurons with somata $A$ and $B$ is then given by

$$\rho = 0.5 \, \mathrm{ms} + (\mathrm{dist}(A, x) + \mathrm{dist}(B, x)) \, v, \tag{8}$$

where v = 2 ms/cm. This delay varies between 0.5 and 1.5 ms and is constant for a given pair of neurons as observed by *Anderson (1985)*.

We assume that neurons in the MNN are randomly placed on the subumbrellar surface. The orientation $\phi$ of their neurites relative to a straight line from the center of the bell to an (arbitrary) rhopalium is drawn from a von Mises distribution, with parameters depending on the position of the neuron,

$$f(\phi | d, \alpha) = \frac{e^{8(d - 0.5) \cos(\phi - 3\alpha)}}{2\pi I_0(8(d - 0.5))}. \tag{9}$$

Here, $d$ is the distance of the neuron from the center (in cm) and $\alpha$ is its polar angle relative to the line from the center to the rhopalium. $I_0(k) = \sum_{m=0}^{\infty} \frac{1}{m! \Gamma(m+1)} \left(\frac{k}{2}\right)^{2m}$ is the modified Bessel function of order zero, normalizing the expression. *Equation (9)* implements the position dependence of the orientation distribution reported in *Aurelia aurita* (*Horridge, 1954a*), by (i) changing the variance of orientations with $d$ and (ii) changing the mean of the orientation distribution with $\alpha$. For comparison, we also consider networks with randomly uniform neurite orientation. *Figure 18* displays example networks with the two different types of orientation distributions.

## Diffuse nerve net

We model the DNN similarly to the MNN, since little is known about it. In particular, we assume the same channel dynamics for DNN as for MNN neurons. There are, however, three main differences between the network models: First, the DNN extends into the bell margin (*Horridge, 1956*), which we take into account by increasing the maximum distance of the neurons from the center of the bell by 0.25 cm (blue hatched area in *Figure 17*). Second, we set the overall length of DNN neurons to

**Table 2.** Synapse model parameters.

| Variable | Value | Unit |
|---|---|---|
| $g_{\mathrm{syn}}$ | 75 | nS |
| $\tau_{\mathrm{rise}}$ | 20 | ms |
| $\tau_{\mathrm{fast}}$ | 3 | ms |
| $\tau_{\mathrm{slow}}$ | 6 | ms |
| $a$ | $9.57 \cdot 10^{-1}$ | |
| $E_{\mathrm{syn}}$ | 4.32 | mV |

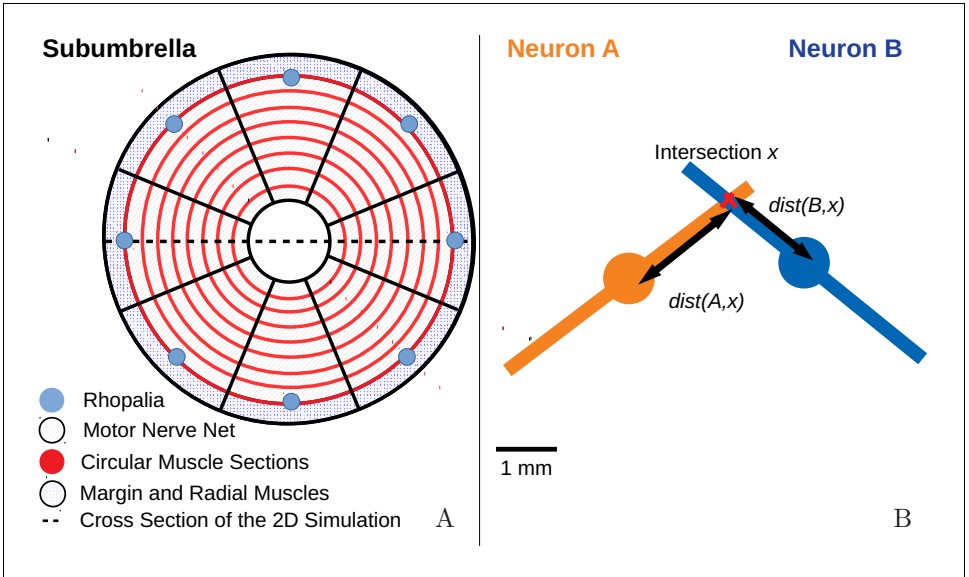

**Figure 17.** The jellyfish model. (**A**) We model the jellyfish subumbrella as a disc with radius 2.25 cm. The MNN somata are embedded in an annulus with an outer radius of 2 cm and an inner radius of 0.5 cm (gray hatched), leaving the margin and the manubrium region void. We assume that the circular swim muscles (thick red) form discrete sections of concentric circles around the manubrium. The centers of these sections are aligned with the positions of the rhopalia. The DNN is distributed over the annulus between manubrium and margin and the margin with width 0.25 cm (blue hatched). For the hydrodynamics simulations, we use a cross-section of the jellyfish as indicated by the dashed line. (**B**) We model the spatial geometry of MNN neurites as line segments (rods) and assume that the soma is in their center (discs). Two neurons are synaptically connected if their neurites overlap. The transmission delay is a function of the distances between the somata and the intersection of their line segments (*Equation (8)*).

2 mm, in agreement with experimental observations (*Passano and Passano, 1971*). Third, neurite orientations are drawn from a uniform distribution. *Figure 19* shows an example DNN network.

## Muscles

To model the activation of circular swim muscles by MNN neurons (see *Figure 17*), we follow a simple model for muscle force twitches used in *Raikova and Aladjov (2002)* and *Contessa and De*

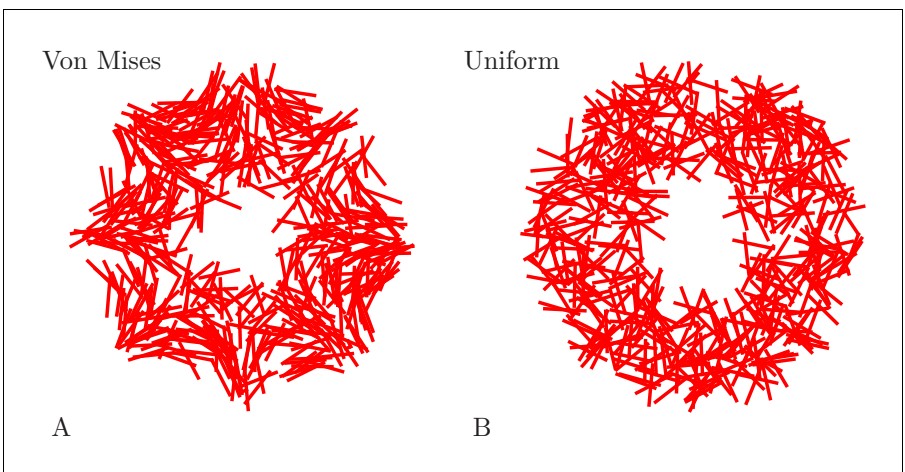

**Figure 18.** Example MNN models. Two MNNs consisting of 500 neurons with von Mises (**A**) or uniformly distributed (**B**) neurite orientation.

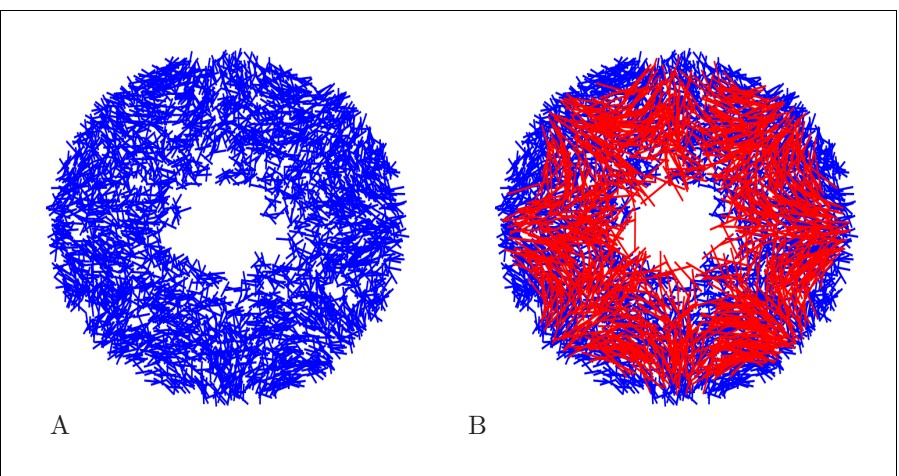

**Figure 19.** Example DNN model. (**A**) A DNN with 3500 Neurons. (**B**) The DNN (blue) and an MNN with 1000 neurons (red) displayed together. The DNN extends further into the bell margin.

*Luca (2013)*: We assume that the time course of a muscle activation evoked by a single spike of an MNN neuron is given by

$$a(t) = t^m e^{-kt} \Theta(t). \tag{10}$$

We choose the rise and relaxation time parameters $m$ and $k$ such that the muscle activity duration is in the range of a variety of jellyfish species (see *Satterlie, 2015*; Table 2 ).

The force exerted by an activated muscle depends on its instantaneous extension. This effect prevents pathological muscle contraction by limiting the range of muscle activity. To incorporate the dependence, we adopt a simple model for force-length relationships (*Battista et al., 2015*), assuming that the maximal force $F_I^j$ that a muscle fiber $j \in \{1, \ldots, 64\}$ of length $L_F^j$ can exert, is given by

$$F_I^j(L_F^j) = F_O \exp\left[ -\left( \frac{L_F^j/L_O^j - 1}{S} \right)^2 \right]. \tag{11}$$

Here, $L_O^j$ is the optimal length, $F_O$ the maximal force, which is generated at length $L_O^j$, and $S$ is a muscle-specific constant. $L_O^j$ is set to the length of the resting muscle. For simplicity, we do not include a force-velocity dependence in our model.

In summary, the force of a muscle fiber $j$ with length $L_t^j$ at time $t$ is in our model

$$f_j(t, L_t) = F_I^j(L_t^j) \sum_{i=0}^{n_j} a\left(t - t_i^j\right), \tag{12a}$$

where $t_0^j, t_1^j \ldots, t_{n_j}^j$ are the spike times of the MNN neurons innervating muscle $j$. We choose the constant $F_O$ such that

$$\max_{t,j} F_O \sum_{i=0}^{n_j} a\left(t - t_i^j\right) = F_{\text{Norm}} \tag{12b}$$

after simulating the nerve net activity. Hence, the muscle strength lies between 0 and $F_{\text{Norm}}$ after an excitation wave has passed through the MNN. All muscles are normalized in the same way, such that the relative strength between them stays constant independent of the number of neurons and the conduction speed.

The circular muscles of *Aurelia aurita* are modeled as blocks of eight muscle units ordered radially in the area of each rhopalium. In total, we thus have 64 muscles (see *Figure 17*). We assume that a neuron is connected to one of those muscles if its somatic position lies in the area covered by the muscle.

The radial muscles in the bell margin are modeled in the same manner as the circular ones. They are separated into eight blocks in the bell margin (see *Figure 17*) and are innervated by DNN neurons in the same way as the circular muscles are innervated by MNN neurons. Their activity is also governed by *Equation (10)- (12)* and they are also normalized in the same manner, independently of the circular muscles. The parameters of the muscle model can be found in *Table 3*.

## Simulation of the swimming motion

### The Immersed Boundary method

To model the swimming behavior of the jellyfish we use the Immersed Boundary (IB) method (*Peskin, 1972*; *Peskin, 2002*). It was originally formulated to study flow patterns surrounding heart valves and has since been used for systems with intermediate Reynolds numbers,

$$\mathrm{Re} = \frac{\rho V L}{\mu},$$

(13)

of $10^{-1}$ to $10^3$. Here, $\rho$ and $\mu$ are the density and the viscosity of the surrounding fluid and $V$ and $L$ are the characteristic velocity and length of the problem (*Battista et al., 2017a*). In our simulations, we set the maximal Reynolds number to approximately 250 by adjusting the viscosity of the fluid. This is in the range of Reynolds numbers calculated for swimming oblate Medusozoans (*Colin and Costello, 2002*) and yields a stable swimming motion in 2D simulations (*Herschlag and Miller, 2011*). We use the IB2D package by *Battista et al. (2015)*, *Battista et al. (2017a)* and *Battista et al. (2017b)* to implement the simulation. The parameters of the IB2D simulations can be found in *Table 4*.

### 2D jellyfish geometry

For our hydrodynamics simulations, we develop a simple 2D construct, which is similarly shaped as 2D geometrical sections of *Aurelia aurita* measured by *Bajcar et al. (2009)* and *McHenry and Jed (2003)*. Our method of defining outlines allows in principle to create a wide variety of shapes including realistic cross sections of both prolate and oblate jellyfish while requiring only few parameters.

We define the relaxed shape of the subumbrella cross-section with length $2r$ by a series of $N_p$ vertices tracing a curve, on each half of the jellyfish. Specifically, the vertices are placed at constant distances $r/N_p$ from one another; the negative angle $\varphi(i)$ between horizontal line and connection of $i^{th}$ and $(i+1)^{th}$ vertex (see *Figure 20*) decreases on the right hand side half with $i = 0, ..., N_p - 1$ as

$$\varphi(i) = -\alpha(1-p)\left(\frac{i}{N_p}\right)^{n_1} - \alpha p\left(\frac{i}{N_p}\right)^{n_2}.$$

(14)

Here, $\alpha$ (usually $\pi/2$) is the angle between the current orientation (center line) of the jellyfish and the horizontal line. The exponents $n_1$ and $n_2$ characterize the jellyfish's curvatures: the higher their values, the more oblate the jellyfish. $p$, a number between 0 and 1, characterizes the contribution of the two curvatures. To preserve the distance between the vertices, the first vertex is placed at half the usual distance (i.e. $r/(2N_p)$) from the center of the subumbrella curve. Analogous expressions hold for the left hand side half. We note that for $n_1 = n_2 = 1$ the subumbrella is a semicircle with radius $2r/\pi$.

The exumbrellar surface is defined by a series of vertices perpendicular to the subumbrella vertices (see *Figure 20*). Specifically, the $i^{th}$ exumbrellar vertex, $i = 1, ..., N_p - 1$, lies at a distance $h(i)$ to

**Table 3.** Muscle model parameters.

| Variable | Value | Unit |
|---|---|---|
| $m$ | 1.075 | |
| $k$ | $2.15 \cdot 10^{-2}$ | |
| $S$ | 0.4 | |
| $F_{\mathrm{Norm}}$ for circular muscles | 0.4 | N |
| $F_{\mathrm{Norm}}$ for radial muscles | 0.8 | N |

**Table 4.** Fluid Simulation parameters.

| Variable | Value | Unit |
|---|---|---|
| $\mu$ | 0.005 | Ns/m$^2$ |
| $\rho$ | 1000 | kg/m$^2$ |
| Time step | 10$^{-5}$ | s |
| x-length of Eulerian grid | 0.06 | m |
| y-length of Eulerian grid | 0.08 | m |
| x-grid size | 180 | |
| y-grid size | 240 | |

the $i^{th}$ subumbrellar vertex, perpendicular to the curve traced by the subumbrellar vertices. We model the height $h(i)$ of the jellyfish umbrella by base height plus a Gaussian hump

$$h(i) = C_{\text{base}}(N_p - i) + C_{\text{amp}} \exp\left(\frac{i^2}{\sigma^2}\right), \tag{15}$$

where $C_{\text{base}}$ is the minimal height of the umbrella and $C_{\text{amp}}$ and $\sigma$ characterize the maximum height and the width of the umbrella's central hump. The parameters used to describe the 2D sections can be found in *Table 5*.

## 2D elastic structure

The jellyfish is an elastic structure filled with fluid; in particular the opening after a swimming contraction is a passive process (*Alexander, 1964*; *Gladfelter, 1972*; *Gladfelter, 1973*). To incorporate this, we also construct the 2D cross-section of the bell as an elastic structure filled with fluid (*Alexander, 1964*): a set of damped springs run across the exumbrellar and the subumbrellar surfaces and connect the two surfaces defined by the vertices of the 2D cross-section (see *Figure 20*). In the IB2D

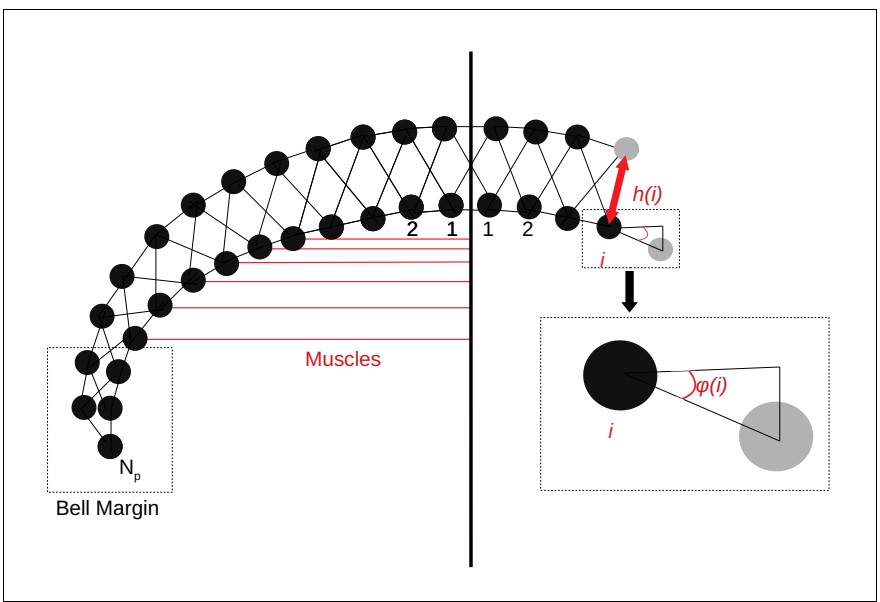

**Figure 20.** The Jellyfish 2D sectional model. The 2D structure consists of two rows of vertices, which are connected by damped springs (black lines). The placement of the vertices in the subumbrella (bottom row) depends only on the angle $\varphi(i)$ (*Equation (14)*). The vertices in the exumbrella (top row) are placed at a distance $h(i)$ (*Equation (15)*) perpendicular to the curve traced by the bottom vertices. The circular muscles (red lines), which contract the bell, create a force (*Equation (12)*) toward the imaginary center line of the jellyfish. No circular muscles are present at the center of the bell and the bell margin.

**Table 5.** Geometry parameters.

| Variable | Value | Unit |
| --- | --- | --- |
| $\alpha$ | $\pi/2$ | |
| $p$ | 0.5 | |
| $N_p$ | 224 | |
| $n_1$ | 1 | |
| $n_2$ | 2 | |
| $C_{\text{base}}$ | 0.5 | mm |
| $C_{\text{amp}}$ | 6 | mm |
| $\sigma$ | 3000 | |
| $k_s^{\text{surface}}$ | $2 \cdot 10^7$ | N/m |
| $k_s^{\text{internal}}$ | $8 \cdot 10^7$ | N/m |
| $b_s$ | 2.5 | kg/s |

package, the force on two vertices with coordinate vectors $X_1, X_2$ connected by a damped spring is defined by

$$\boldsymbol{F}_{\text{s}} = k_s \left( 1 - \frac{R_L}{||\boldsymbol{X_1} - \boldsymbol{X_2}||} \right) (\boldsymbol{X_1} - \boldsymbol{X_2}) + b_s \frac{\text{d}}{\text{d}t} ||\boldsymbol{X_1} - \boldsymbol{X_2}||, \tag{16}$$

where $R_L$ is the resting length, $k_S$ the spring stiffness and $b_S$ the damping coefficient.

Since the length of the 3D circular muscles and their radius are proportional, we model them by muscles that are attached at subumbrellar vertices and exert the forces given by *Equation (12)* directly toward the center line (see *Figure 20*, red). To simulate the contraction of the radial muscles, we place DNN innervated muscles between neighboring vertices alongside the subumbrellar springs of the bell margin.

## Acknowledgements

We thank Peter AV Anderson and Alexander P Hoover for fruitful discussions, Jonas Böhm for providing the photo of the jellyfish and the German Federal Ministry of Education and Research (BMBF) for support via the Bernstein Network (Bernstein Award 2014, 01GQ1710). FP is also supported by the SMARTSTART Joint Training Program of the Bernstein Network and the VolkswagenStiftung.

## Additional information

### Funding

| Funder | Grant reference number | Author |
| --- | --- | --- |
| Federal Ministry of Education and Research | 01GQ1710 | Fabian Pallasdies Sven Goedeke Wilhelm Braun Raoul Memmesheimer |
| SMARTSTART Joint Training Program of the Bernstein Network and the VolkswagenStiftung | SmartStart2 | Fabian Pallasdies |

The funders had no role in study design, data collection and interpretation, or the decision to submit the work for publication.

### Author contributions

Fabian Pallasdies, Conceptualization, Software, Formal analysis, Validation, Investigation, Visualization, Methodology; Sven Goedeke, Wilhelm Braun, Conceptualization, Supervision, Validation,

Methodology; Raoul-Martin Memmesheimer, Conceptualization, Supervision, Funding acquisition, Validation, Methodology, Project administration

### Author ORCIDs
Fabian Pallasdies (iD) https://orcid.org/0000-0001-5359-4699
Sven Goedeke (iD) https://orcid.org/0000-0001-5314-345X
Wilhelm Braun (iD) http://orcid.org/0000-0002-9419-3311

### Decision letter and Author response
Decision letter https://doi.org/10.7554/eLife.50084.sa1
Author response https://doi.org/10.7554/eLife.50084.sa2

## Additional files

### Supplementary files
• Transparent reporting form

### Data availability
No experimental data sets were generated in this study. Simulation parameters for all figures can be found in the manuscript and its supplements. Hydrodynamics simulations were performed with the IB2D package by Nicholas A Battista (https://github.com/nickabattista/ib2d).

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
