## [Decision Letter]

**Acceptance summary:**

This paper develops a computational model of a jellyfish that integrates multiple
scales from single-neuron activity to neuronal networks to muscle activation and
fluid-structure interaction all coupled together to produce locomotion and behavior.
The modeling is impressive and makes experimentally testable predictions. The
results on the effect of the network size on the travel distance and biases in the
direction of locomotion are interesting and indicate a good match of biological
network size (efficiency) with effective locomotion. The role of the rhopalia and
the interplay between Diffuse Nerve Net and Motor Nerve Net in the production of
turning behavior is very informative and interesting. The work is very well done,
and the authors have addressed all concerns raised in the first review. In
particular, they are to be commended for doing the additional modeling with
Romanes-type cuts thereby rooting the model more firmly in the biological data.

**Decision letter after peer review:**

Thank you for submitting your article "From single neurons to behavior in the
jellyfish Aurelia aurita" for consideration by *eLife*. Your
article has been reviewed by two peer reviewers, and the evaluation has been
overseen by Ronald Calabrese as the Senior and Reviewing Editor. The following
individuals involved in review of your submission have agreed to reveal their
identity: Richard Satterlie (Reviewer #1); Eva Kanso (Reviewer #2).

The reviewers have discussed the reviews with one another and the Reviewing Editor
has drafted this decision to help you prepare a revised submission.

Essential revisions:

The full reviewer comments are provided and should be fully addressed, but as a guide
to the revision, we emphasize a few points brought out in these reviews.

1) In the comparison of uniform MNN versus von Mises MNN, it will strengthen the
paper significantly if the authors provide a superposition of the kind of
experimental cuts Romanes conducted on the two models to see if/how they survive
conduction around these interdigitating barriers.

2) There are published figures of Aurelia nerve nets using immunohistochemical
techniques (both MNN and DNN) (Figure 5 in Biological Bulletin 226: 29-40 (2014))
where information could be gleaned regarding neurite orientation and density for
comparison with the models and the resulting estimates.

3) The manuscript itself must be improved to make it more accessible to the research
communities working on underwater locomotion and behavior, and to broaden the impact
of this work beyond the computational neuroscience community.

*Reviewer #1:*

This is an interesting manuscript that uses modeling of established properties of
neurons and neuronal connections to examine the connection between neuronal
conduction and behavior, including biomechanics. Modeling is an appropriate way to
approach this problem due to difficulties in making multiple recording from
scyphozoan neurons. Also, Aurelia is a good choice since it has been used for
behavioral and neurobiological projects since the time of Romanes.

The writing is clear and concise, and the figures are appropriate. I am not a
modeler, so I have mostly minor comments on the manuscript, with a few major
comments.

In the comparison of uniform MNN versus von Mises MNN, I would like to have seen a
superposition of the kind of experimental cuts Romanes conducted on the two models
to see if they would survive conduction around these interdigitating barriers.

Also, there are published figures of Aurelia nerve nets using immunohistochemical
techniques (both MNN and DNN) where information could be gleaned regarding neurite
orientation and density for comparison with the models and the resulting
estimates.

Not applicable to this manuscript: I would like to encourage the authors to use their
models to investigate conduction in colonial nerve nets of anthozoan corals since
Horridge and Anderson found that there were species-specific differences with some
exhibiting through-conduction of polyp withdrawal while others had a restricted area
of polyp withdrawal. Of the latter, some exhibited incremental conduction while
other exhibited decremental conduction. Modeling could elucidate properties of nerve
nets that might produce these differences (it would also be a further test of the
models).

*Reviewer #2:*

The paper proposes a computational model of jellyfish that integrates multiple scales
from single-neuron activity to neuronal networks to muscle activation and
fluid-structure interaction all coupled together to produce locomotion and behavior.
The modeling is impressive! I find the results on the effect of the size of the
neural network on the travel distance and direction of locomotion very interesting,
as well as the results on how the interplay between DNN and MNN could produce
turning behavior. However, I have a few comments and suggestions for improving the
manuscript.

1) After reading the manuscript, I remained unsure what are the mathematical
differences between the MNN and the DNN models and how the two networks are coupled
together. I suggest that the authors use a schematic figure that clearly shows the
different components of the model and how they are all coupled together. The
schematic should be placed in the main document, upfront, not at the end of the
manuscript in the Materials and methods section. For inspiration on how to produce
such a schematic, consult the animal locomotion literature. For example, see Figure
2 of Dickinson's 2000 Science paper "how animals move: an integrative
view" and elaborate on the different components of the nervous system of the
jellyfish, how it couples to the musculoskeletal system and to the external fluid
environment.

2) The coupling between the multiple scales from single neuron all the way to
biomechanics and behavior is impressive. The power of such mathematical model lies
in the ability to use it to make predictions and testable hypotheses on how behavior
is enacted at the physiological level, which the authors touch upon in the Results
section. However, the literate review in the Introduction is heavily biased towards
describing the jellyfish nervous system, and refers to behavior very briefly, if at
all. In fact, the literature review on jellyfish turning behavior is introduced in
the Results section, which in my opinion is not effective. I suggest that all
relevant background information, including jellyfish behavior be introduced in the
Introduction, together with clearly stating existing or new hypotheses and research
questions, such as "how do jellyfish turn away to avoid undesired
stimuli", this is only mentioned at the end of the Results section.

3) From a biomechanics standpoint, mathematical models have been developed by Lisa
Fauci, Eric Tytell, and collaborators in the context anguilliform swimming, where
muscle activation is coupled to fluid-structure interactions to produce swimming
behavior, with body undulations not prescribed a priori, but emerging from the
coupling between the internal dynamics (muscle activation) and fluid-structure
interactions. To me, the novelty of the present work is in modeling the neural
network itself. However, I feel that it is important for the authors to acknowledge
the work done in the aquatic locomotion community on related problems, such as in
the context of undulatory swimming. This should be done in the Introduction section.
In the least, such connection will serve to attract wider audience to this work.

4) Figure 7 is supposed to "validate" or "benchmark" the model
against experimental study. I suggest that the authors show the experimental results
from Figure 2 of McHenry and Jed, 2003, in Figure 7. On the same note, I'm also
almost confident that many groups have worked on fluid-structure interaction
simulations showing the vortex ring around a symmetric jellyfish. I suggest that the
authors properly reference this body of literature, and use it to both benchmark
their flow simulation results and to distinguish their model from existing models
that assume prescribed bell deformations.

In sum, I think the mathematical model and the main results are potentially exciting,
but the manuscript itself could be improved to make them more accessible to the
research communities working on underwater locomotion and behavior, and to broaden
the impact of this work beyond the computational neuroscience community.

---

## [Author Response]

Essential revisions:The full reviewer comments are provided and should be fully addressed, but as a
guide to the revision, we emphasize a few points brought out in these
reviews.1) In the comparison of uniform MNN versus von Mises MNN, it will strengthen the
paper significantly if the authors provide a superposition of the kind of
experimental cuts Romanes conducted on the two models to see if/how they survive
conduction around these interdigitating barriers.

We tested both the radial and the circular cutting experiments performed by Romanes
in the two types of networks (von Mises and uniform). We find that the
through-conducting property is preserved even for severe cuts. We added one
subsection and two new figures (Figures 7 and 8) to the Results which describe these
simulations.

2) There are published figures of Aurelia nerve nets using immunohistochemical
techniques (both MNN and DNN) (Figure 5 in Biological Bulletin 226: 29-40
(2014)) where information could be gleaned regarding neurite orientation and
density for comparison with the models and the resulting estimates.

We added two paragraphs to the Discussion, highlighting that our model suggests a low
density of the DNN, since this is beneficial for efficient turning. Consistent with
this, the immunohistochemical staining experiments displayed in Satterlie and
Eichinger, 2014, find a much lower DNN than MNN neuron density. We further
incorporated the experimental finding of general radial bias in the DNN neurite
orientation, and find that a radial bias leads to a slightly lower propagation speed
(see Figure 20 in the revised manuscript). We have attempted to estimate the neuron
densities of the MNN and DNN from Figure 3 in Satterlie and Eichinger, 2014. Direct
extrapolation to adult medusa sizes as considered in our manuscript leads, however,
to excessively large neuron numbers. We discuss this in more detail in our response
below.

3) The manuscript itself must be improved to make it more accessible to the
research communities working on underwater locomotion and behavior, and to
broaden the impact of this work beyond the computational neuroscience
community.

To improve the accessibility of our manuscript and broaden its appeal beyond the
computational neuroscience community, we added an introductory figure including a
schematic overview of our model to the beginning of the manuscript (Figure 1 in the
revised manuscript). We also added another subsection to the Introduction describing
in more detail previous approaches towards modeling of underwater locomotion and
summarizing what is known about jellyfish swimming. We also expanded our account of
previous work on fluidstructure interactions and vortex ring dynamics in the Results
section of the manuscript.

Reviewer #1:This is an interesting manuscript that uses modeling of established properties of
neurons and neuronal connections to examine the connection between neuronal
conduction and behavior, including biomechanics. Modeling is an appropriate way
to approach this problem due to difficulties in making multiple recording from
scyphozoan neurons. Also, Aurelia is a good choice since it has been used for
behavioral and neurobiological projects since the time of Romanes.

We thank the reviewer for his careful reviewing of our manuscript and his inspiring
and encouraging suggestions. In the following we address his comments one by
one.

The writing is clear and concise, and the figures are appropriate. I am not a
modeler, so I have mostly minor comments on the manuscript, with a few major
comments.In the comparison of uniform MNN versus von Mises MNN, I would like to have seen
a superposition of the kind of experimental cuts Romanes conducted on the two
models to see if they would survive conduction around these interdigitating
barriers.

Following the suggestion of the referee, we tested both the radial and the circular
cutting experiments performed by Romanes in both von Mises and uniform networks. In
agreement with the experiments the simulations show that conduction survives even
for severe cuts. There is no qualitative difference between von Mises and uniform
networks. We describe our simulations and our findings in the newly added subsection
“Cutting experiments" in the Results section of the revised manuscript. It
includes two new figures, Figures 7 and 8 and two additional animations.

Also, there are published figures of Aurelia nerve nets using immunohistochemical
techniques (both MNN and DNN) where information could be gleaned regarding
neurite orientation and density for comparison with the models and the resulting
estimates.

We added two paragraphs in the Discussion comparing the results of Satterlie and
Eichinger, 2014, on neuron orientation and density with our model. Since Satterlie
and Eichinger discovered a general radial bias in the neurite orientation of DNN
neurons we tested if this bias has a similar effect on the propagation speed as our
von Mises model of the MNN. We find that a radial bias implemented with a von Mises
distribution leads to a slightly slower propagation speed (Figure 20 in the revised
manuscript). We discuss this in the Discussion section as well. We also highlight
that the lower density of the DNN in comparison to the MNN allows the jellyfish to
better steer its swimming motion, since a lower density leads to a slower conduction
speed. This means that the delays between MNN and DNN can be larger. We were,
however, unable to obtain a reliable estimate for the total number of neurons in DNN
and MNN networks for the medusa sizes used in our manuscript. Concretely, we
attempted to estimate the neuron density of the MNN using Figure 3A in Satterlie and
Eichinger, 2014. A very coarse (lower) estimate of 25 neurites in the shown area of
about 0.025 mm2 leads to a density of 100,000 neurons per cm2, which would result in
an excessively large number of neurons in adult medusa. In the literature we found
that for Hydra the total number of neurons was estimated as 5600 (Bode et al.,
1973). For Cubozoa, the neuron number in the ring nerve was estimated to lie between
8500 and 17,000 (Garm et al., 2007). A partial remedy could follow from taking into
account that the MNN neurites are longer than the side length of the area shown in
Figure 3 of Satterlie and Eichinger, 2014. Then a single neurite contributes to the
density by crossing several areas as the one shown. But to compute the resulting
density we need to assume a typical neurite length, which is unknown in Figure 3A
and may vary with medusa size, which is not given in the article. Further, we
identify about 25 somas in Figure 3B of Satterlie and Eichinger, 2014, which would
again result in an excessively large number of neurons for the DNN in adult medusa.
Taken together, it is unclear to us if and how the observed neuron densities in
Figure 3 from Satterlie and Eichinger can be extrapolated to the medusa sizes
considered in our manuscript. Therefore, we think that a quantitative comparison is
not adequate at this point.

Not applicable to this manuscript: I would like to encourage the authors to use
their models to investigate conduction in colonial nerve nets of anthozoan
corals since Horridge and Anderson found that there were species-specific
differences with some exhibiting through-conduction of polyp withdrawal while
others had a restricted area of polyp withdrawal. Of the latter, some exhibited
incremental conduction while other exhibited decremental conduction. Modeling
could elucidate properties of nerve nets that might produce these differences
(it would also be a further test of the models).

Thank you very much for highlighting this interesting possible further application
and test of our modeling approach to us.

Reviewer #2:The paper proposes a computational model of jellyfish that integrates multiple
scales from single-neuron activity to neuronal networks to muscle activation and
fluid-structure interaction all coupled together to produce locomotion and
behavior. The modeling is impressive! I find the results on the effect of the
size of the neural network on the travel distance and direction of locomotion
very interesting, as well as the results on how the interplay between DNN and
MNN could produce turning behavior. However, I have a few comments and
suggestions for improving the manuscript.1) After reading the manuscript, I remained unsure what are the mathematical
differences between the MNN and the DNN models and how the two networks are
coupled together. I suggest that the authors use a schematic figure that clearly
shows the different components of the model and how they are all coupled
together. The schematic should be placed in the main document, upfront, not at
the end of the manuscript in the Materials and methods section. For inspiration
on how to produce such a schematic, consult the animal locomotion literature.
For example, see Figure 2 of Dickinson's 2000 Science paper "how animals
move: an integrative view" and elaborate on the different components of the
nervous system of the jellyfish, how it couples to the musculoskeletal system
and to the external fluid environment.

Following your suggestion we have added a new schematic figure to the Introduction of
the manuscript (Figure 1 in the revised manuscript). In this schematic figure, we
explain the extent of the two nerve nets, how they are coupled to the rhopalia and
which muscle groups they innervate. Inspired by your comment we have also added an
explanation of the DNN setup in the Results subsection “The Mechanism of
Turning" such that the reader does not have to go back and forth between
Results and Materials and methods to understand the setup.

2) The coupling between the multiple scales from single neuron all the way to
biomechanics and behavior is impressive. The power of such mathematical model
lies in the ability to use it to make predictions and testable hypotheses on how
behavior is enacted at the physiological level, which the authors touch upon in
the Results section. However, the literate review in the Introduction is heavily
biased towards describing the jellyfish nervous system, and refers to behavior
very briefly, if at all. In fact, the literature review on jellyfish turning
behavior is introduced in the Results section, which in my opinion is not
effective. I suggest that all relevant background information, including
jellyfish behavior be introduced in the Introduction, together with clearly
stating existing or new hypotheses and research questions, such as "how do
jellyfish turn away to avoid undesired stimuli", this is only mentioned at
the end of the Results section.

Following your suggestion, we added the subsection “The Hydrodynamics of
Swimming" on previous studies of jellyfish swimming and turning to the
Introduction. This subsection more comprehensively reviews the literature on
jellyfish swimming in both experimental and theoretical studies and highlights
current open questions.

3) From a biomechanics standpoint, mathematical models have been developed by
Lisa Fauci, Eric Tytell, and collaborators in the context anguilliform swimming,
where muscle activation is coupled to fluid-structure interactions to produce
swimming behavior, with body undulations not prescribed a priori, but emerging
from the coupling between the internal dynamics (muscle activation) and
fluid-structure interactions. To me, the novelty of the present work is in
modeling the neural network itself. However, I feel that it is important for the
authors to acknowledge the work done in the aquatic locomotion community on
related problems, such as in the context of undulatory swimming. This should be
done in the Introduction section. In the least, such connection will serve to
attract wider audience to this work.

In the newly added subsection “The Hydrodynamics of Swimming" of the
Introduction we acknowledge the work of the aquatic locomotion community, in
particular the work by Fauci and Tytell and we discuss how our study relates to
it.

4) Figure 7 is supposed to "validate" or "benchmark" the
model against experimental study. I suggest that the authors show the
experimental results from Figure 2 of McHenry and Jed, 2003, in Figure 7. On the
same note, I'm also almost confident that many groups have worked on
fluid-structure interaction simulations showing the vortex ring around a
symmetric jellyfish. I suggest that the authors properly reference this body of
literature, and use it to both benchmark their flow simulation results and to
distinguish their model from existing models that assume prescribed bell
deformations.

We included the results from McHenry and Jed, 2003, into our Figure 10 and expanded
the comparison in the results. We now also show that the agreement between the
recorded data and our simulations can be improved by changing parameters of the
jellyfish model's geometry. We more comprehensively discuss the potentials and
shortcomings of our 2D simulation in this context and compare it with other
jellyfish simulations. Furthermore we expanded our description of the simulated
swimming motion to better compare our results to other simulations and recordings of
jellyfish swimming.

In sum, I think the mathematical model and the main results are potentially
exciting, but the manuscript itself could be improved to make them more
accessible to the research communities working on underwater locomotion and
behavior, and to broaden the impact of this work beyond the computational
neuroscience community.

Thank you again for your helpful comments. We have carefully addressed them and
improved the manuscript, in particular with respect to the listed points.